

# Monitoring aerosols over Europe: an assessment of the potential benefit of assimilating the VIS04 measurements from the future MTG/FCI geostationary imager

Maxence Descheemaecker[1], Matthieu Plu[1], Virginie Marécal[1], Marine Claeyman[2], Francis Olivier[2], Youva Aoun[3], Philippe Blanc[3], Lucien Wald[3], Jonathan Guth[1], Bojan Sič[1], Jérôme Vidot[4], Andrea Piacentini[5], Béatrice Josse[1]

1CNRM, Météo-France - CNRS, UMR3589, 31057, Toulouse, France
2Thales Alenia Space, Cannes la Bocca, 06156, France
3Mines ParisTech, PSL Research University, O.I.E.- Center for Observation, Impacts, Energy, Sophia Antipolis, 06904, France
4Centre de Météorologie Spatiale, Météo-France, 22300, Lannion, France
5CERFACS – CNRS, UMR5318, 31057, Toulouse, France

*Correspondence to*: Maxence Descheemaecker (maxence.descheemaecker@meteo.fr)

**Abstract.** The study assesses the possible benefit of assimilating Aerosol Optical Depth (AOD) from the future spaceborne sensor FCI (Flexible Combined Imager) for air quality monitoring in Europe. An Observing System Simulation Experiment (OSSE) was designed and applied over a 4-months period that includes a severe pollution episode. The study focuses on the FCI channel centred at 444 nm, which is the shortest wavelength of FCI. A Nature Run (NR) and four different Control Runs of the MOCAGE chemistry-transport model were designed and evaluated to guarantee the robustness of the OSSE results. The AOD synthetic observations from the NR were disturbed by errors that are typical of the FCI. The variance of the FCI AOD at 444 nm was deduced from a global sensitivity analysis that took into account the aerosol type, surface reflectance and different atmospheric optical properties. The experiments show a general benefit on all statistical indicators of the assimilation of the FCI AOD at 444 nm for aerosol concentrations at surface over Europe, and also a positive impact during the severe pollution event. The simulations with data assimilation reproduced spatial and temporal patterns of PM$_{10}$ concentrations at surface better than without assimilation all along the simulations and especially during the pollution event. This work demonstrates the capability of data from the future FCI sensor to bring an added value to the MOCAGE aerosol simulations, and in general, to other chemistry transport models.

## 1 Introduction

Aerosols are liquid and solid compounds suspended in the atmosphere, whose sizes range from a few nanometers to several tens of micrometers, and whose lifetime in the troposphere varies from a few hours to a few weeks (Seinfeld and Pandis, 1998).





Stable sulfate aerosols at high altitude can last for years (Chazette et al., 1995). The sources of aerosols may be natural (dusts, sea salt, ashes from volcanic eruptions, for instance) or anthropogenic (from road traffic, residential heating, industries, for instance), and they can be transported up to thousands of kilometers. Aerosols are known to have significant impacts on climate (IPCC, 2007) and on air quality and further on human health as WHO (2014) estimated over 3 million deaths in 2012 to be

due to aerosols.

Aerosols absorb and diffuse solar radiation, which lead to local heating of the aerosol layer and cooling of the climate system through the backscatter of solar radiation to space for most of the aerosols, except for black carbon (Stocker et al., 2013). The absorption of solar radiation modifies the vertical temperature profile, affecting the stability of the atmosphere and cloud formation (Seinfeld and Pandis, 1998). Aerosols, as condensation nuclei, play a significant role in the formation and life cycle

of clouds (Seinfeld and Pandis, 1998). Deposition of aerosols on Earth's surface may also affect surface properties and albedo. All these effects show that aerosols play a key role on the energy budget of the climate system.

Aerosols, also called particulate matter in the context of air quality, are responsible for serious health problems all over the world, as they are known to favor respiratory and cardiovascular diseases as well as cancers (Brook et al., 2004). The World Health Organization (WHO) has set regulatory limits for aerosol concentrations, respectively 20 µg.m$^{-3}$ and 10 µg.m$^{-3}$ annual

mean for PM$_{10}$ and PM$_{2.5}$ (particulate matter with a diameter less than 10 and 2.5 µm, respectively) concentrations. The presence of a dense layer of aerosols can also affect air traffic by the reduction of visibility (Bäumer et al., 2008) and by risks of disruptions of engines of air planes (Guffanti et al., 2010). Therefore, it is essential to accurately determine the evolution of the concentration and size of the different types of aerosols in space and time, in order to assess their effect on climate and on air quality and to mitigate their impacts. A pertinent approach to achieve a continuous and accurate monitoring of aerosols is

to combine measurements and models, a good example being the Copernicus Atmosphere Monitoring Service (CAMS) (Eskes et al., 2015).

Ground-based stations, which measure aerosol and gas concentrations in-situ, have been used for several decades to monitor air quality, such as the stations in the Air Quality e-Reporting program (AQeR, https://www.eea.europa.eu/data-and-maps/data/aqereporting-2) from the European Environment Agency (EEA). Other observations can also be used to measure

aerosols. The AERONET (AErosol RObotic NETwork) program (https://aeronet.gsfc.nasa.gov/) performs the retrieval of the Aerosol Optical Depth (AOD) at several ground stations (Holben et al., 1998). Similarly, AOD observations can be retrieved from images taken in different channels by imagers aboard Low Earth Orbit (LEO) or GEOstationary (GEO) satellites. Generally, AOD from satellite provides a better spatial coverage than ground-based stations at the expense of additional sources of uncertainty, such as the surface reflectance. An example of AOD product from LEO satellites is the Daily Level 2

AOD, from the Moderate Resolution Imaging Spectroradiometer (MODIS) (Levy et al., 2013) sensor on board Terra and Aqua (MOD 04 & MYD 04 products). This AOD product is provided at a 10 km resolution every 5 min. Sensors on geostationary orbit satellites can continuously scan one third of Earth's surface much more frequently than low Earth orbit satellites. The SEVIRI (Spinning Enhanced Visible and Infra-Red Imager) sensor, aboard MSG (Meteosat Second Generation), is an example of a GEO sensor providing information on aerosols. Different AOD are retrieved over lands from SEVIRI data in the VIS0.6





and VIS0.8 channels, respectively centered at 0.635 µm (0.56 µm – 0.71 µm) and 0.81 µm (0.74 µm – 0.88µm). AOD products are retrieved following different methods. Carrer et al. (2010) presented a method to estimate a daily quality-controlled AOD based on a directional and temporal analysis of SEVIRI observations of channel VIS0.6. Another method consists in matching simulated Top Of the Atmosphere (TOA) reflectances (from a set of 5 models) with TOA SEVIRI reflectances (Bernard et al.,

2011) to obtain an AOD for VIS0.6. Another method (Mei et al., 2012) estimates the AOD and the aerosol type by analysing the reflectances at 0.6 and 0.8 µm in three orderly scan times. These methods derive AOD for specific channels, from the combined analysis of several channels and very often using several images if not all of a day to have information.

Numerical models, even if they are subject to errors, are necessary to describe the variability of the aerosol types and of their

concentrations with space and time, as a complement to observations. Aerosol forecasts on regional and global scales are made by three-dimensional models, such as the chemistry-transport model (CTM) MOCAGE (Sič et al., 2015; Guth et al., 2016). MOCAGE is currently used daily to provide air quality forecasts to the French platform Prev'Air (Rouil et al., 2009) and also to the European CAMS ensemble (Marécal et al., 2015). Data assimilation of AOD can be used in order to improve the representation of aerosols within the model simulations (Sič et al, 2016). Studies on geostationary sensors have also proved a

positive effect of the assimilation of AOD, see e.g. Yumimoto, et al. (2016), who assessed this positive effect using the AOD at 550 nm from AHI (Advanced Himawari Imager) sensor aboard Himawari-8. Other studies have shown the positive impact of assimilating synthetic aerosol data from future geostationary instruments in preparation of new satellite sensors. Timmermans et al (2009) presented the results of assimilating synthetic AOD together with ground-based PM$_{2.5}$ measurements and showed a positive impact on the estimation of surface PM$_{2.5}$ concentrations. Claeyman et al. (2011) also used an observing

systems simulation experiments (OSSE) approach to evaluate the benefit of geostationary instruments to monitor gas pollutant concentrations in the lowermost troposphere.

The future geostationary Flexible Combined Imager (FCI, URD Eumetsat, 2010), that will be aboard the Meteosat Third Generation satellite (MTG), will perform a full disk in 10 min, and in 2.5 min for the European Regional-Rapid-Scan which

covers one-quarter of the full disk, with a spatial resolution of 1 km at nadir and around 2 km in Europe. Like AHI, FCI is designed to have multiple wavelengths and the assimilation of its data into models should be beneficial to aerosol monitoring. The aim of the paper is to assess the possible benefit of assimilating measurements from the future MTG/FCI sensor for monitoring aerosols over Europe. This study focuses on the assimilation of AOD of the channel VIS04 (centered at 444 nm), as the channel VIS04 is new compared to the present SEVIRI sensor, and besides, a short wavelength is expected to be more

accurate to detect small particles, such as PM$_{2.5}$., than longer wavelengths (Petty, 2006). As this sensor is not yet operational, an OSSE approach (Timmermans et al., 2015) is used in this study to assess its ability to improve the aerosols analysis, by assimilation of 444 nm AOD. The general principle of OSSEs (Observing System Simulation Experiments) is to assess the added value of future sensors by assimilating synthetic observations in different simulations. AOD synthetic observations from FCI VIS04 will be assimilated in MOCAGE.



The experimental setup is explained in Sect. 2. It contains a description of the OSSE, of the MOCAGE model, and its assimilation system. Then, the case study and an evaluation of the ability of the reference simulation to represent a true state of the atmosphere are presented. The calculation of synthetic observations is explained in Sect. 3. An evaluation of the control

simulations is made in Sect. 4. In Sect. 5, the results of the assimilation of FCI synthetic observations are presented and discussed. Finally, Sect. 6 concludes this study.

## 2 Methodology

### 2.1 Experimental setup

Figure 1 shows the general principle of the OSSE (Timmermans et al, 2015). A reference simulation, called "Nature Run"

(NR) is assumed to represent the "true" state of the atmosphere. AOD synthetic observations are generated from the addition of AOD retrieved from the NR and the error characteristics of FCI. These error characteristics are described in Sect. 3. The second kind of simulations in the OSSE is the "Control Run" (CR) simulation. The differences between NR's output and CR's output should approximate those between a state-of-the-art model and the actual atmosphere. Finally, the assimilation run (AR) is done by assimilation in the CR of the synthetic observations. To assess the added value of the instrument, a comparison

is made between the output of the AR and the NR and between the CR and the NR. If the AR is closer to the NR than the CR, it means that the observations provide useful information to the assimilation system. The differences between AR and CR quantify the added value of the instrument.

The NR should be as close as possible to the actual atmosphere because it serves as the reference to produce the synthetic

observations. The temporal and spatial variations of the NR should approximate those of actual observations. An evaluation of the NR, presented in Sect. 2.2, includes a comparison of the model with aerosol concentrations and AOD data from ground-based stations.

In addition, the differences between the NR and the CR must be significant and approximate those between the CR and the actual observations. Ideally, the NR and CR should be run with different models, as the use of the same model could lead to

over-optimistic results (Masutani et al., 2010); this issue is called the "identical twin" problem. In the case of using the same model, it is strongly recommended to evaluate the spatio-temporal variability of the NR and its differences with the CR to avoid this "identical twin" problem (Timmermans et al., 2015). As MOCAGE is both used for NR and CR in the present study, a method similar to that used in Claeyman et al. (2011) is proposed. Instead of one CR, various CR simulations (Fig. 1) are performed in different configurations, and they are assessed independently and compared to the NR to ensure the robustness

of the OSSE results. An evaluation of those differences is presented in Sect. 4.



## 2.2 MOCAGE

The CTM model used in this study is MOCAGE (Modèle de Chimie Atmosphérique à Grande Echelle, Guth et al, 2016), that has been developed for operational and research purposes. MOCAGE is a three-dimensional model that covers the global scale, down to regional scale using two-way nested grids. MOCAGE vertical resolution is not uniform: the model has 47

vertical sigma-hybrid altitude-pressure levels from the surface up to 5 hPa. Levels are denser near the surface, with a resolution of about 40 m in the lower troposphere and 800 m in the lower stratosphere.

MOCAGE simulates gases (Josse et al., 2004; Dufour et al., 2004), primary aerosols (Martet et al., 2009; Sič et al., 2015) and secondary inorganic aerosols (Guth et al, 2016). Aerosols species in the model are primary species: desert dust, sea salt, black carbon and organic carbon, and secondary inorganic species: sulfate, nitrate and ammonium, formed from gaseous precursors

in the model. For all types of aerosols, the same 6 bin sizes are used between 2 nm and 50 µm: 2 nm -10 nm - 100 nm - 1 µm - 2.5 µm - 10 µm - 50 µm. All emitted species are injected every 15 mins in the five lower levels (up to 0.5 km), following an hyperbolic decay with altitude: the fraction of pollutants emitted in the lowest level is 52 %, and then respectively 26 %, 13 %, 6 % and 3 % in the four levels above. Such a vertical repartition ensures continuous concentration fields in the first levels, which guarantee a proper behaviour of the of the semi-Lagrangian advection scheme. Carbonaceous particles are emitted using

emission inventories. Sea salt emissions are simulated using a semi-empirical source function (Gong, 2003; Jaeglé et al., 2011) with the wind speed and the water temperature as input. Desert dust are emitted, using wind speed, soil moisture and surface characteristics based on Marticorena and Bergametti (1995) which give the total emission mass, that is then distributed in each bin according to Alfaro et al. (1998). Secondary inorganic aerosols are included in MOCAGE using the module ISORROPIA II (Fountoukis and Nenes, 2007), which solves the thermodynamic equilibrium between gaseous, liquid and solid compounds.

Chemical species are transformed by the RACMOBUS scheme, which is a combination of the RACM scheme (Regional Atmospheric Chemistry Mechanism; Stockwell et al., 1997) and the REPROBUS scheme (Reactive Processes Ruling the Ozone Budget in the Statosphere; Lefèvre et al., 1994). Dry and wet depositions of gaseous and particulate compounds are made as in Guth et al. (2016).

MOCAGE uses meteorological forecasts (wind, pressure, temperature, specific humidity, precipitation) as input, such as

Météo-France operational meteorological forecast from ARPEGE (Action de Recherche Petite Echelle Grande Echelle), or ECMWF (European Centre for Medium-Range Weather Forecasts) meteorological forecast from IFS (Integrated Forecast System). A semi-langrangian advection scheme (Williamson and Rasch, 1989; Staniforth and Côté, 1991), a parameterization for convection (Bechtold et al., 2001) and a diffusion scheme (Louis, 1979) are used to transport gaseous and particulate species.



### 2.3 Assimilation system PALM

The assimilation system of MOCAGE (Massart et al., 2005; 2009), is based on the 3-Dimensional First Guess at Appropriate Time (3D-FGAT) algorithm. This method consists of minimizing the cost function $J$:

$$J(\delta x) = J_b(\delta x) + J_o(\delta x) = \frac{1}{2}(\delta x)^T \mathbf{B}^{-1}\delta x + \frac{1}{2}\sum_{i=0}^{N}(d_i - \mathbf{H}_i\delta x)^T \mathbf{R}_i^{-1}(d_i - \mathbf{H}_i\delta x),$$ (1)

where $J_b$ and $J_o$ are respectively the part of the cost function related to the model background and to the observations; $\delta x = x - x^b$ is the difference between the model background $x^b$ and the state of the system $x$; $d_i = y_i - H_i\,x^b(t_i)$ is the difference between the observation $y_i$ and the background $x^b$ in the observations space at time $t_i$; $H_i$ is the observation operator; $\mathbf{H}$ its linearized version; $\mathbf{B}$ is the background covariance matrix; and $\mathbf{R_i}$ is the observation covariance matrix at time $t_i$.

The general principal for the assimilation of AOD is the same as in Sič et al. (2016). The control variable $x$ used in the minimization is the 3D total aerosol concentration. After minimization of the cost function, an analysis increment $\delta x^a$, is obtained, which is a 3D-total aerosol concentration. This increment $\delta x^a$ is then converted into all MOCAGE aerosol bins according to their local fractions of the total aerosol mass in the model background. The result is added to the background aerosol field at the beginning of the cycle. Then the model is run over the 1-hour cycle length to obtain the analysis. The state at the end of this cycle is used as a departure point for the background model run of the next cycle.

The observation operator $\mathbf{H}$ for AOD uses as input the concentrations of all bins (6) of the seven types of aerosols and the associated optical properties. For this computation also, the control variable $x$ is converted into all MOCAGE aerosol bins according to their local fractions of the total aerosol mass in the model background. The AOD is computed for each model layer to obtain, by summing, the AOD of the total column. This computation is based on Wiscombe's Mie code scheme for spherical and homogeneous particles (Wiscombe, 1980, 1979, revised 1996). For every MOCAGE aerosol species, different refractive indices are used. They are issued from Kirchstetter et al. (2004) for organic carbon and from the Global Aerosol Data Set (GADS, Köpke et al., 1997) for other aerosol species. The hygroscopicity of sea salts and secondary inorganic aerosols are taken into account based on Gerber (1985).

### 2.4 Case study

The period extends from the 1st of January to the 30th of April 2014, and includes several days of PM pollution over Europe. From the 7th to 15th of March, a secondary particles episode (EEA report 2014) occurs, while from 29th March to 5th April a dust plume originating from the Sahara Desert propagates Northwards to Europe (Vieno et al., 2016).

The MOCAGE simulation covers the whole period from January to April 2014, on a global domain at 2° resolution and a nested regional domain, that covers Europe, from 28 °N to 72 °N and from 26 °W to 46 °E, at 0.2 ° resolution (see Fig. 2). A 4-month spin-up is made before the simulation. The NR is forced by ARPEGE meteorological analysis. Emissions of chemical species in the global domain come from MACCity (van der Werf et al., 2006; Lamarque et al., 2010; Granier et al., 2011; Diehl et al., 2012) for anthropogenic gas species and biogenic species are from GEIA for the global and regional domain.





ACCMIP project emissions are used for anthropogenic organic and black carbon emissions at the global scale. The TNO-MACC-III inventory for year 2011 provides anthropogenic emissions in the regional domain. TNO-MACC-III emissions are the latest update of the TNO-MACC inventory based on the methodology developed in the MACC-II project described in Kuenen et al. (2014). These anthropogenic emissions are completed, on our regional domain, at the boundary of the MACC-

5    III inventory domain by emissions from MACCity. Daily biomass burning sources of organic and black carbon and gases from the Global Fire Assimilation System (GFAS) (Kaiser et al., 2012) are injected in the model. To enhance its realism, the NR has been improved, by introducing secondary organic aerosols (SOA) into the model, to well fit the observations made at ground-based stations over Europe. Standard ratios from observations (Castro et al., 1998) are used to simulate the portion of secondary carbon species, 40 % in winter, from the primary carbon species in the emission input.

The NR is compared to real observations from AERONET AOD observations and AQeR surface concentrations, using several statistical indicators. The Mean Bias (B), defined as:

$$B = \frac{1}{N}\sum_{i=1}^{N}(f_i - o_i),\tag{2}$$

where $N$ is the number of pairs $(f_i\,;o_i)$, $f_i$ is the forecast value and $o_i$ is the observation value.

The modified normalized bias (MNMB) is defined as:

$$MNMB = \frac{2}{N}\sum_{i=1}^{N}\frac{f_i - o_i}{f_i + o_i},\tag{4}$$

and varies between - 2 and 2.

The Root Mean Square Error (RMSE) is defined as:

$$RMSE = \sqrt{\frac{1}{N}\sum_{i=1}^{N}(f_i - o_i)^2}\,,\tag{3}$$

As the RMSE is dominated by high residuals due to squaring operation, the Fractional Gross Error (FGE) is used in this study. It is defined as:

$$FGE = \frac{2}{N}\sum_{i=1}^{N}\frac{|f_i - o_i|}{f_i + o_i},\tag{5}$$

and varies between 0 and 2.

Another indicator is the factor of 2 (FactOf2), which represents the fraction of the forecast dataset ranged within a factor of 2

from the observation dataset.

The Pearson correlation coefficient $(R_p)$ is often used to measure the extent to which patterns in the forecast dataset match those in the observation dataset in a linear aspect. It ranges from -1 to 1 and has the following formula:

$$R_P = \frac{\sum_{i=1}^{N}(f_i - \bar{f})(o_i - \bar{o})}{N\sigma_f\sigma_o},\tag{6}$$

where $\bar{f}$ and $\bar{o}$ are the means of the datasets and $\sigma_f$ and $\sigma_o$ are the standard deviations:





$$\sigma_f = \sqrt{\frac{1}{N}\sum_{i=1}^{N}(f_i - \bar{f})^2} \ ; \ \sigma_o = \sqrt{\frac{1}{N}\sum_{i=1}^{N}(o_i - \bar{o})^2} \qquad , \qquad (7)$$

The Spearman correlation is a mean to assess a monotonic relationship between the datasets. If $rg$ denotes the rank of a value, it is defined as:

$$R_S = \frac{\sum_{i=1}^{N}(rg(f_i) - \overline{rg(f)})(rg(o_i) - \overline{rg(o)})}{N\sigma_{rg(f)}\sigma_{rg(o)}}, \qquad (8)$$

The AQeR stations are mainly located over Western Europe (Fig. 2). 597 and 535 stations are respectively used for the $PM_{10}$, $PM_{2.5}$ comparison. Stations, classified as 1 to 5, have been selected following Joly and Peuch (2012). This selection keeps stations that are representative of background air pollution, which is the range of scale that the model and the satellite may represent. Figure 2 represents the mean surface concentration of the NR and selected AQeR measurements over the domain,

from January to April 2014. The left panel shows the $PM_{10}$ concentrations of the NR in the background and the AQeR concentrations as circle, while the right panel shows the $PM_{2.5}$ concentrations. The concentration of the NR $PM_{10}$ and $PM_{2.5}$ are generally a bit underestimated compared to observations. Nevertheless, on both figures, the spatial variability and particularly the maxima are reasonably well represented. Over the European continent, the NR and AQeR data show clear maxima in the center of Europe for both $PM_{10}$ and $PM_{2.5}$ concentrations, even if the NR underestimates these maxima.

Table 1 shows the statistical indicators of this comparison for hourly surface concentrations in $PM_{10}$ and $PM_{2.5}$. A negative mean bias is observed, around -6.23 µg.m$^{-3}$ (~ -35.1 %) for $PM_{10}$ and -3.20 µg.m$^{-3}$ (~ -24.7 %) for $PM_{2.5}$. The RMSE is equal to 16.2 µg.m$^{-3}$ for $PM_{10}$ and 11.9 µg.m$^{-3}$ for $PM_{2.5}$ while the FGE equals to 0.56 and 0.543. The factor of two is equal to 64.7 % and 67.5 % for $PM_{10}$ and $PM_{2.5}$. Pearson and Spearman correlations are respectively 0.452 and 0.535 for $PM_{10}$ and $PM_{2.5}$ and 0.537 and 0.602 for $PM_{10}$ and $PM_{2.5}$. The NR underestimation is greater for $PM_{10}$ than for $PM_{2.5}$ in relative differences.

This suggests a lack of aerosol concentrations in the $PM_{10-2.5}$ (concentration of aerosols between 2.5 µm and 10 µm). Not taking into account wind-blown crustal aerosols may cause a potential underestimation of PM in models (Im et al., 2015). Taking them into account needs a detailed ground type inventory to compute those emissions unavailable in MOCAGE. For $PM_{2.5}$, the underestimation of aerosol concentrations can be due to a lack of carbonaceous species (Prank et al., 2016).

A time-series graph of the median NR surface concentrations and the median surface concentrations of the AQeR stations are

25 presented in Fig. 3. Compared to ground-based AQeR data (in black), the NR (in purple) generally underestimates the $PM_{10}$ and the $PM_{2.5}$ concentrations, especially during the 7th-15th March pollution episode. However, the variations and maxima of the NR concentrations of PM are generally well represented. Furthermore, around 65 % of model concentrations are relatively close to observations as shown by the factor of 2 in Table 1. The variability of NR concentrations is thus consistent with AQeR station concentrations.

Table 2 gives an evaluation of the NR against the daily mean of the AOD at 500 nm obtained from 84 AERONET stations in the regional domain from January to April 2014. The statistical indicators show good consistency between the NR and AERONET observations, even if the NR tends to overestimate AOD. The uncertainty of the vertical distribution of aerosol



concentrations in the NR may explain this overestimation. The overestimation may also be due to uncertainties of the most important parameters in the computation of the AOD for each level layer, namely, the refractive index, density, hygroscopicity and the mixing state. The mixing state is a property of particles to be arranged in different chemical configurations across the particle population. As mentioned above, particles in MOCAGE are separated in individual species and in 6 bins and assumed

to be spherical and homogeneous. Curci et al. (2015) studied the impact of the refractive index, density, hygroscopicity and the mixing state on simulated optical properties by comparing observations to models from the Air Quality Model Evaluation International Initiative (AQMEII-2). An estimation of uncertainties due to the assumption of the mixing state is calculated at 30-35 % on simulated AOD, while uncertainties from other parameters are around 10 %. Those uncertainties could explain the overestimation of the AOD from the NR, even if the surface concentration of the $PM_{10}$ of the NR is underestimated. Another

explanation of the overestimation of AOD is an overestimation of particulate matters larger than 10 µm (and therefore not include in the $PM_{10}$ comparison) which cause higher AOD.

As a result, the NR simulation exhibits surface concentrations and AOD in the same range compared to those from ground-based stations and shows similar spatial and temporal variations, which makes the NR acceptable for the OSSE.

**3 Generation of synthetic AOD observations**

Synthetic observations are created over the MOCAGE simulated regional domain from the 3D fields of the NR variables: primary organic and black carbon concentrations, desert dust concentrations, sea salt concentrations, secondary inorganic aerosol concentrations (ammonium, nitrate and sulfate), relative humidity, temperature and pressure. From these fields, AODs are calculated for the FCI VIS04 channel with the same computation module for aerosol optical properties as the observation operator. An AOD error is introduced using characterize errors of the FCI. To characterize the error in the channel VIS04, the

simulator developed by Aoun (2016, Aoun et al., 2015), based on the Radiative Transfer Model (RTM) libRadtran (Mayer and Killings, 2005), has been used.

This simulator computes the radiance measured (Fig.4, Step 1) by the different spectral bands of FCI, under different atmospheric conditions (total column water vapour, ozone contents, aerosol species concentrations and spectral optical depths),

ground albedos and solar zenithal angles. It takes into account the spectral response sensitivity and signal-to-noise ratio of each FCI spectral band. In order to build a statistical error model of AOD, a Monte-Carlo method has been implemented by operating the FCI simulator on a large number of atmospheric profiles.

Since the FCI response may depend on the aerosol types, independent model errors per OPAC (Optical Properties of Aerosols and Clouds, Hess et al., 1998) types have been computed: dust, maritime clean, maritime polluted, continental clean,

continental average, continental polluted, and urban. For each OPAC type, the distribution of the atmospheric conditions has been built from a MOCAGE simulation ran over the whole 2013 year. Aerosol species described in OPAC are mineral dusts (nuc. mode: MINM; acc. mode: MIAM; coa. mode: MICM), sea salts (acc. mode: SSAM; coa. mode: SSCM), soluble aerosols





(water soluble: WASO), insoluble aerosols (INSO) and soot aerosols (SOOT). A correspondence of aerosol species has been made between MOCAGE and OPAC (Table 3) as in Ceamanos et al. (2014). Then some criteria are applied on the vertical distributions of aerosol species and their concentrations (Table 4) to classify the MOCAGE profiles into the different OPAC types. A large number of profiles have not been classified because they did not fit into the criteria. The aim of this step is to produce the most accurate and realistic histograms of AOD used for the Monte-Carlo based simulations of FCI radiances.

For each spectral band and OPAC aerosol types, and under the hypothesis of statistical independence of each variable, the simulator has been used to compute radiance for a Monte-Carlo-based random draws of 200.000 sets of inputs, following histograms. The Fig. 4 (Step 1) presents an example of histograms of the inputs and the corresponding histogram of the resulting radiances for the spectral band VIS04 (415-475 nm), for the aerosol type "continental clean". A measurement noise (white Gaussian noise) has been added to the simulated radiance, following a specific model of signal-to-noise ratio (i.e. standard deviation of noise over the resulting radiance). Then the radiances were converted in reflectance by dividing it by the corresponding TOA radiances including the cosine of the solar zenithal angle.

A global sensitivity analysis (GSA) has been performed using the Sobol indices (Fig. 4, step 2) of order one and two (Sobol, 1990; 1993) based on the ANOVA decomposition (Hoeffding, 1948). Following Sobol (1996), the GSA can be used to determine the key inputs but also to determine a truncated version of the Hoeffding (ANOVA) functional decomposition, with key inputs, that approximates the analysed reflectance. In the case of the simulations with 5 inputs, the Hoeffding decomposition up to the order 2 comprises 15 functions (5 for each single parameters and 5x4/2 = 10 functions for each couple). Under the assumption of independent inputs, the Sobol indices enable a ranking of inputs or couple of inputs with respect their variance-based importance in the total output variance. The Fig. 5 presents the ranking of the Sobol indices of orders 1 and 2 for the output reflectance of VIS04 for the continental clean aerosol type. The blue bars represent the Sobol indices in descending order of single or couple of inputs. The red line represents the cumulated sum of the sorted Sobol indices. The variability of the solar zenithal angle, the ground albedo and the AOD are the three largest Sobol indices in that order and, together, they are at the origin of more than 98 % of the total variance of the output reflectance. For the example of VIS04 with the aerosol type "continental clean", the reflectance $R$ can be approximated by the following equation:

$$R = f_3(\theta_S) + f_2(\rho_g) + f_1(\tau) + \epsilon, \tag{9}$$

where $f_1$, $f_2$, and $f_3$ are functions, $\theta_S$ is the solar zenith angle, $\rho_g$ is the ground albedo and $\tau$ is the aerosol optical depth. The approximation error $\epsilon$, also called the modelling error, exhibits a root mean square (RMS) less than 0.7 W m$^{-2}$ sr$^{-1}$ µm$^{-1}$ (1.5 % of the mean radiance values of 47.3 W m$^{-2}$ sr$^{-1}$ µm$^{-1}$). For all OPAC groups, the dependence of reflectance for VIS04 on the total ozone column and water vapour is negligible and is not taken into account in the reflectance approximation.

In this approximation, it is then possible to isolate the AOD $\tau$ with respect the measured reflectance $R$, the other key inputs but the AOD. In the case of VIS04/continental clean, these remaining inputs are $\theta_S$, $\rho_g$ and the approximation error $\epsilon$:





$$\tau = F\left(R, \theta_S, \rho_g, \epsilon\right), \tag{10}$$

The reflectance $R$ is associated with a measurement noise. Except for the solar zenithal angle, the other inputs may be known with a given uncertainty. In the AOD retrieval a main source of uncertainty is due to surface albedo. The surface albedo fields are retrieved from MODIS using the Radiative Transfer Model RTTOV (Vidot et al., 2014), and the relative error of albedo

used to create synthetic observations of the study is 10 %. The other inputs, such as total ozone column, comes from the NR simulation. With (Eq. 10), it is then possible to run a new Monte-Carlo analysis (Fig. 4, step 3) with random draws of noise measurements, approximation error $\epsilon$ and uncertainty on the key inputs, in order to assess the root mean square error (RMSE) of the AOD retrieval. As an example, Fig. 6 presents the estimated RMSE of the AOD retrieval from the VIS04 reflectance, for continental clean aerosol type, for an uncertainty of 5 % on the ground albedo. The RMSE depends both of the solar zenithal

angle and on the ground albedo.

To create the synthetic observations, each NR profile of aerosol is associated to an OPAC type (Table 5) using three parameters: the surface concentration, the main surface species and the proportion in relation to the total aerosols concentrations. A species is described as a main species if its concentrations, [species], is above each other concentrations, for example DD is a main

species if [DD]>[SS] & [DD]>[IWS]. An example of NR profiles (7[th] March 2014 at 12 UTC) decomposed in OPAC type is presented in Fig. 7. On every profiles, an AOD error is introduced, by addition of a unbiases Gaussian with standard deviation equal to the RMSE, depending on the OPAC type. In order to keep a large number of profiles, the criteria are less restrictive than the ones described in Table 4. By this process, a few portions of the profiles will be dismissed, such as profiles over ocean where *IWS* (Insoluble, Water Soluble and Soot; Table 5) is greater than DD (Desert Dust) and SS (Sea Salt). Furthermore,

night-time profiles and cloudy profiles are also dismissed. An example of the synthetic observations is presented in Fig. 8. It represents the NR AOD, the synthetic observations and the relative error for the 7[th] March 2014 at 12 UTC. After these filters apply, between 10 % and 20 % of profiles are kept every hour. Figure 9 represents the average number of profiles, that are available for assimilation, retained per day. The density of these profiles is denser in the south of the domain. This is directly correlated to the quantity of direct sunlight available. Over the continent, between 1 and 4 profiles can be assimilated per day.

## 4 Controls runs (CRs) and their comparison to NR

Sect. 2 showed an evaluation of the NR compared to real observations. Another requirement of the OSSE is the evaluation of differences between the NR and the CR. Various CR simulations have been performed to evaluate the behaviour of the OSSE on different CR configurations and prove its robustness. The NR and CRs use different setups of MOCAGE. The CRs use IFS

meteorological forcings, while the NR uses ARPEGE meteorological forcings. The use of different meteorological inputs is expected to yield differences in the transport of pollutant species, and changes in dynamic emissions of sea salt and desert dust. To introduce more differences between the CRs and NR, changes in the emissions are also introduced.





Table 6 indicates the changes made on the different model parameters to create 4 distinct CR simulations. The first control run, CR1, uses the same inputs as the NR except for the meteorological forcings. Other control runs (CR2, CR3, CR4) do not have the (SOA) improvement made in the NR (Sect. 2) and CR1 simulations. Finally, CR3 and CR4 change from other simulations by different vertical repartitions of emissions in the five lowest levels. In CR3, the pollutants are emitted with a

slowest decay with height than the NR (with repartition from 30 % at surface and respectively 24 %; 19 %; 15 %; 12 % for the four levels above), and in the CR4 emissions are only injected in the lowest level. These changes aim to generate simulations that are more more significantly different from the NR than the first two control runs.

The four CRs are compared to the NR for $PM_{10}$ and $PM_{2.5}$ surface concentration considering virtual observations located at the same locations as the AQeR stations. A time-series of daily means of surface concentrations at simulated stations is

presented in Figure 10, for NR and CRs simulations from the 1st January to the 30th April 2014. The $PM_{10}$ concentrations of the NR (in purple) are mostly greater than the $PM_{10}$ concentrations in the CRs. During the period of late March and early April (around the 90th day of simulation) the NR concentrations of $PM_{10}$ are close to those of the CR2, CR3 and CR4, and less than those of the CR1 by about few $\mu gm^{-3}$. In terms of $PM_{2.5}$, the CRs concentrations are also underestimating the NR concentration. As for $PM_{10}$, around the 90th day of simulation, the concentrations of CR1 are above the concentrations of the NR.

These tendencies can also be observed in Figure 11, which represents a scatter plot of CRs concentrations as a function of NR concentrations for the daily means of surface concentration in $PM_{10}$ and $PM_{2.5}$ at the virtual stations. The CR1 concentrations are fairly close to those of the NR concentrations with a coefficient of regression about 0.801 and 0.835 for $PM_{10}$ and $PM_{2.5}$. Other CRs underestimate the NR concentrations. This tendency is stronger for $PM_{10}$ than for $PM_{2.5}$. The regression coefficient of the CR2, CR3, CR4 are respectively 0.596, 0.583 and 0.607 for $PM_{10}$ and 0.570, 0.505 0.647 for $PM_{2.5}$. For both $PM_{10}$ and

$PM_{2.5}$ concentrations, the underestimation is more important for high values of the NR concentrations than for low values.

The statistical indicators in Tables 7 and 8 are consistent with Fig. 10 and 11. The CR1 is close to the NR with a bias of -1.3 (-8.2 %) $\mu g.m^{-3}$ for $PM_{10}$ and -0.8 (-6.2 %) $\mu g.m^{-3}$ for $PM_{2.5}$. CR4 bias is around -2.9 (-20.5 %) $\mu g.m^{-3}$ for $PM_{10}$ and -1.8 (-15.1 %) $\mu g.m^{-3}$ for $PM_{2.5}$. The two other CRs highly underestimate $PM_{10}$ and $PM_{2.5}$ concentrations with a bias of -4.5 $\mu g.m^{-3}$ (-35.2 %) and -3.9 $\mu g.m^{-3}$ (-37.4 %) respectively for CR2 and -4.8 $\mu g.m^{-3}$ (-38.1 %) and -4.4 $\mu g.m^{-3}$ (-42.6 %) for the CR3.

These biases are in agreement with the literature. Prank et al. (2016) measure a bias around -5.8 for $PM_{10}$ and -4.4 $\mu g.m^{-3}$ for $PM_{2.5}$ for the median of four CTMs against ground-based stations in winter. In Marécal et al. (2015), statistical indicators for an ensemble of seven models are presented for winter. A bias between -3 and -7 $\mu g.m^{-3}$ is observed for the median ensemble. The PM concentrations of our CRs compared to the NR are characteristic of models compared to observations.

Prank et al. (2016) also show other indicators for the median of models, such as the temporal correlation and the factor of 2.

Their correlations are around 0.7 for $PM_{2.5}$ and 0.6 for $PM_{10}$ and are close to those for our CRs simulations that vary from 0.644 to 0.732 for $PM_{2.5}$ and from 0.572 to 0.671 for $PM_{10}$. Their factor of 2 equals 65 % for $PM_{10}$ and 67 % for $PM_{2.5}$. The factor of 2 of the CRs ranges between 70 % and 90 % for both $PM_{10}$ and $PM_{2.5}$ concentrations. The RMSE of CRs simulations ranges from 8 $\mu g.m^{-3}$ to 10 $\mu g.m^{-3}$ for $PM_{10}$ concentrations, which is slightly under the RMSE of the ensemble from the study of Marécal et al. (2015) which ranges between 10 and 15 $\mu g.m^{-3}$. The FGE of the study of Marécal et al. is equal to 0.55, while



the FGE of CRs varies from 0.33 to 0.51. Our CRs simulations slightly underestimate the model relative error. Thus, compared to literature, the CRs (especially the CR4) are different enough from the NR to be representative of state-of-the-art simulations. Between the CRs and the NR there are important spatial differences in surface concentrations of PM, as demonstrated in Figure 12 for $PM_{10}$. The relative differences, Pearson correlation and the FGE are represented in this figure. Over the Atlantic Ocean,

the CRs concentrations are relatively close to the NR, except for the CR4 which overestimates the concentration of $PM_{10}$. All CRs present high concentrations of $PM_{10}$ all over North Africa. This corresponds to high emissions of desert dust over this area, which cause an important overestimation of $PM_{10}$ compared to the NR. This overestimation can also be observed around all the Mediterranean Basin. The CRs tend to overestimate the $PM_{10}$ concentrations over Spain, Italy, the Alps, Greece, Turkey, the north of the UK, the Iceland and the Norway. The overestimation over the Alps, Iceland and Norway are located at places

of negligible concentrations. Over the rest of the European continent, CRs underestimate the concentration of $PM_{10}$, slightly for CR1, but very pronounced for CR2, CR3, and especially for CR4. The area where the consistency between the CRs and the NR is better is the Atlantic Ocean with a correlation ranging from 0.6 to 0.9 and a low FGE around 0.3. Over the Mediterranean Basin the correlation varies significantly between 0 and 1. Low correlations correspond to high FGE around 1. Over the continent, the correlation varies from 0.4 to 0.9 following a west-east axis. The correlations are slightly greater for

CR1 than for the other CRs. The FGE over the continent changes significantly between the CR1 and the other CRs, respectively around 0.35 and 0.55. Similar conclusions can be obtained for the $PM_{2.5}$ comparison (see complementary materials). A similar comparison has been done for the AOD between the CRs simulations and the NR simulation (see complementary materials). In summary, the control runs present spatial variability along with temporal variability. The closest CR to the NR is the CR1. In terms of surface concentrations in PM, the CR3 is the most distant, while in terms of AOD the CR4 is the most distant.

Those differences and the use of different CRs, coupled with the realism of the NR, demonstrate the robustness of the OSSE to evaluate the added value provided by AOD derived from the FCI.

## 5 Assimilation of FCI synthetic observations

The purpose of this paper is to assess the potential contribution of FCI VIS04 channel to the assimilation of aerosols on a

continental scale. In our OSSE, MOCAGE represents the atmosphere with a horizontal resolution of 0.2 ° (around 20 km at the equator). Synthetic observations are therefore computed at the model resolution although FCI scans around 1 km resolution at the equator and 2 km over Europe. To fit with the timestep of our assimilation cycle, synthetic observations are also created every hour, although the future FCI imager could retrieve radiance observations every 10 minutes over the globe, and 2.5 minutes over Europe with the European Regional-Rapid-Scan. This means that for each profile of our simulation, only one

synthetic observation is available each hour, instead of 24x10x10 at best (FCI scans 24 times an hour, with a spatial resolution 10 times higher than the model over the Europe). The use of one observation for each profile in an assimilation window is due to the assimilation system design that does not allow multiple observations for a same profile. In practice, future FCI





observations could be averaged over each MOCAGE profile to reduce the impact of the instrument errors on assimilated observations. Avoiding over-optimistic results is also a reason to only assimilate one observation (non-averaged).

The 3D-FGAT assimilation scheme integrates the synthetic observations described in Sect. 3.

Before assimilation, a screening process is applied to the synthetic observations to keep spatially only 1 pixel out of 4 in order to reach a better convergence of the cost function (not shown), and to reduce the computation time. The spatial correlation length of the B background covariance matrix is set to 0.4° in order to have a spatial impact of the assimilation on the simulation while not having multiple coverage of assimilated observations over one profile. Assimilation simulations (ARs) are run for all CRs simulations using the same generated set of synthetic observations over the period of 4 months, from the 1st of January to the 30th of April. The standard deviation of errors used for B and R matrix are estimated respectively at 24 % and 12 %, as in Sič et al. (2016).

To assess the impact of the assimilation of FCI synthetic AOD observations, the CR forecasts and the AR analyses are compared to the assimilated synthetic observations. Figure 13 shows the histograms of the differences between the synthetic observations and the forecast field (in blue) and between synthetic observations and analyzed fields (in purple) for the four ARs simulations. The histograms follow a Gaussian shape, and the distribution of the analysed values are closer to the synthetic observations than the forecast values. The spread of the histograms is smaller for the analysed fields than for the forecast fields. The assimilation of synthetic AODs hence improved the representation of AOD fields in the assimilation simulations. Besides, the spatial comparisons between the simulations and the NR show great improvements in the AOD fields of simulations by assimilation of the synthetic observations (see supplementary material Fig. S5, S6, S7 and S8). As the increment is applied to all aerosol bins and that $PM_{10}$ corresponds to 5 of the 6 bins while $PM_{2.5}$ to only 4, we expect better corrections for $PM_{10}$ concentrations than for $PM_{2.5}$ concentrations.

To validate the results of the OSSEs, the simulations are compared to the reference simulation (NR) over the period. Figure 14 exhibits the spatial differences in surface concentrations of $PM_{10}$ between the ARs and the NR. It shows the mean relative bias, the correlation and the FGE for every simulations. Using Figure 12 as a reference, the relative bias, the FGE and the correlation have been improved over the domain after assimilation for all simulations. Over the European continent, all simulations show a strong improvement of the statistical indicators, while over North Africa and the Mediterranean Sea the improvement is intermediate. Nevertheless, the mean bias over the ocean tends to increase for the simulations, especially for the simulation 4. This can also be observed for the $PM_{2.5}$ concentration comparison (see supplementary material S1, S2, S3 and S4).

The assimilation of the synthetic observations has a positive impact at each layer of the model. The mean vertical concentrations of $PM_{10}$ and $PM_{2.5}$ of the different simulations are respectively represented in Figure 15 and 16 from the surface (level 47) up to 6 km (level 30). The positive impact along the vertical of the assimilation of AOD in the CTM MOCAGE is



due to the use of the vertical representation of the model to distribute the increment. Sič et al (2016) showed that the assumption of using the vertical representation of the model gives good assimilation results with the regular MOCAGE setup that distributes emissions over the 5 lowest vertical levels. The CRs simulations, in red, overestimate the $PM_{10}$ concentrations of the NR, in purple, due to the overestimation of desert dust concentrations in the CRs simulations. This overestimation is not

present in the $PM_{2.5}$ concentrations because this is the fraction of aerosols where there are few desert dusts. For the first three simulations, the vertical $PM_{10}$ concentrations are well corrected by the assimilation, while for simulation 4, the correction is less relevant for the levels near the surface. The assimilation tends to decrease the $PM_{2.5}$ concentrations above the level 42 and to increase the concentrations under that level. Simulation 4 presents a decay of the surface concentrations of $PM_{2.5}$. The correction of concentrations is more pertinent for the $PM_{10}$ concentrations than for the $PM_{2.5}$ concentrations, which was

expected.

The vertical distribution of aerosol concentrations between the CR4 simulations and the NR explains why the bias over the ocean tends to increase. At the lowest level, the concentration of $PM_{10}$ is more important, since the CR4 emits only at the surface level, while the AOD is less important, since the aerosol loss by dry deposition increases. The positive increment is therefore added preferentially to the surface level, which increases the bias at surface.

To evaluate the capability of the FCI 444 nm channel observations to improve aerosol forecast in an air quality scenario, the ARs simulations have been compared to the NR using the synthetic AQeR stations as in Sect. 4. Tables 9 and 10 show the statistics of the comparison between the ARs and the NR for $PM_{10}$ and $PM_{2.5}$ concentrations. With regard of the comparison of the CRs against the NR in Tables 7 and 8, the ARs are more consistent with the NR. The bias is reduced for both $PM_{10}$ and

$PM_{2.5}$ concentrations. The RMSE and the FGE decrease while the Factor of 2 and the correlations increase for all ARs compared to their respective CRs.

The daily medians of $PM_{10}$ and $PM_{2.5}$ concentrations at all stations are represented over time in Figure 17 and in Figure 18 for the 4 simulations. The assimilation reduces the gap between the simulations and the NR over the entire period. Around the secondary inorganic aerosol episode, 65[th] day of simulation, the improvements of $PM_{10}$ and $PM_{2.5}$ surface concentrations are

significant for simulations 2, 3 and 4.

From an air quality monitoring perspective, the assimilation of the FCI synthetic AOD at 444 nm in MOCAGE improves strongly the surface $PM_{10}$ concentrations in the 4 simulations over the European continent for the period January-April 2014.

To quantify the improvement of simulations through the assimilation of FCI synthetic observations over a high spatial and

temporal episode of pollution for (7[th]-15[th] March) over Europe, maps of relative concentrations of $PM_{10}$ and FGE are respectively represented for the CRs comparison and for the ARs comparison in Figure 19 and in Figure 20. The simulations CR2, CR3 and CR4 underestimate $PM_{10}$ concentrations for 70 % over all Europe compared to the NR. The FGE presents high values going from 0.55 to 0.85. The assimilation of synthetic AOD improves meaningfully the surface concentrations of aerosols over the continent in the simulations, but the simulations still underestimate the $PM_{10}$ concentrations by 30-20 %.





Important changes in the FGE are noticeable, with values dropping from 0.55-0.85 down to 0.2-0.4 for all simulations. Over the other areas, the assimilation reduces significantly the relative bias and the FGE. Thus, the assimilation of synthetic observations improves significantly the representation of the surface $PM_{10}$ concentrations of simulations during the pollution episode.

In summary, the use of synthetic observations at 444 nm of the future sensor FCI through assimilation improves significantly the aerosol fields of the simulations over the European domain from January to April 2014. These improvements are located all over the domain with best results over the European continent and the Mediterranean area. The improvement of the vertical profile of aerosol concentrations is also noticeable. The first two simulations give better results over the ocean than simulations

10   3 and 4, due to a closer representation of the vertical profile of the aerosol concentrations. This may show an overly optimistic aspect of the OSSE of the first two simulations. The simulations lead to sufficiently reliable results since the shapes of their vertical profile of aerosol concentrations are different from those of the NR. These differences are caused by the way emissions are injected in the atmosphere (higher for simulation 3 and lower for simulation 4). The simulations 3 and 4 present robust results over continent, despite the differences in the vertical representation of aerosol concentrations.

## 6 Conclusion

An OSSE method has been developed to quantify the added value of assimilating future MTG/FCI VIS04 AOD (444 nm) in the chemistry and transport model MOCAGE. The characteristic errors of the FCI have been computed from a sensitivity analysis and introduced in the computation of synthetic observations from the NR. An evaluation of the realistic state of the

atmosphere of the NR has been done, as well as a comparison of CR simulations with the NR, in order to avoid the identical twin problem mentioned in Timmermans et al. (2009). Furthermore, different control run simulations have been set up as in Claeyman et al. (2011) to avoid this issue. The results of the OSSE should hence be representative of the results that the assimilation of real retrieved AODs from the FCI sensor will bring.

Although the use of a single synthetic observation per profile and the choice of an albedo error of 10% are pessimistic choices, the assimilation of synthetic AOD at 444 nm showed a positive impact, particularly for the European continental air pollution. The simulations with data assimilation reproduced spatial and temporal patterns of $PM_{10}$ concentrations at surface better than without assimilation all along the simulations and especially during the high pollution event of March. The improvement of analysed fields is also expected for other strong pollution event such as a volcanic ash plume. This capability of synthetic

observations to improve the analysis of aerosols is present for the 4 set of simulations which show the capability of future data from the FCI sensor to bring an added value within the CTM MOCAGE aerosol forecasts, and in general, in atmospheric composition models.



The assimilation has been made using only one observation per model profile. The use of multiple observations using a "super-observation" approach, by spatial and temporal averaging, should reduce the instrumental errors and lead to even better results. Some of the results over ocean showed a breakdown of the concentrations due to the way assimilated information is split between the aerosol bins following the model concentrations on the vertical. The use of multiple wavelengths, using the Ångström exponent, could avoid this problem by better distributing the increment of AOD between the different bins and hence the different species. Sič et al. (2016) also recommended the use of other types of observations, such as lidars, in the assimilation process to introduce information over the vertical.

The results presented here in this OSSE are encouraging for the use of future FCI AOD data within CTMs for the wavelength VIS04 centered at 444 nm. The use of other channels could bring complementary information, such as the channel NIR2.2 that is expected to be less sensitive to fine aerosols but more sensitive to large aerosols such as desert dusts and sea salt aerosols.

## Acknowledgements

We acknowledge that the MTG satellites are being developed under an European Space Agency contract to Thales Alenia Space.

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



**Tables :**

|  | Bias (µg/m3) | RMSE (µg/m3) | FGE | FactOf2 | $R_P$ | $R_S$ |
|---|---|---|---|---|---|---|
| NR's $PM_{10}$ | - 6.23 (~ - 35.1 %) | 16.2 | 0.56 | 64.7 % | 0.452 | 0.537 |
| NR's $PM_{2.5}$ | - 3.20 (~ - 24.7 %) | 11.9 | 0.543 | 67.5 % | 0.535 | 0.602 |

**Table 1: Bias (B), RMSE, FGE, Factor of 2, Pearson correlation (Rp) and Spearman correlation (Rs) of the NR simulation taking as reference the AQeR observations for hourly $PM_{10}$ and $PM_{2.5}$ concentrations from January to April 2014.**

|  | Bias | MNMB | RMSE | FGE | $R_P$ |
|---|---|---|---|---|---|
| NR | 0.043 | 0.39 | 0.09 | 0.531 | 0.56 |

**Table 2: Bias (B), MNMB, RMSE, FGE and Pearson correlation (Rp) between the NR simulation and AERONET station for daily 500 nm AOD from January to April 2014.**

| Species | DD | SS | SIA | OC | BC |
|---|---|---|---|---|---|
| MIAM, MICM, MINM | 1 | - | - | - | - |
| SSAM, SSCM | - | 1 | - | - | - |
| WASO | - | - | 1 | 0.5 | 0.2 |
| INSO | - | - | - | 0.5 | - |
| SOOT | - | - | - | - | 0.8 |

**Table 3: Correspondence of MOCAGE and OPAC aerosol species. Aerosol species of OPAC are mineral dust (nucleation mode: MINM ; accumulation mode: MIAM ; coarse mode: MICM), sea salts (accumulation mode: SSAM ; coarse mode: SSCM), soluble aerosols (water soluble: WASO), insoluble aerosols (INSO) and soots (SOOT), while aerosols of MOCAGE are desert dusts (DD), sea salts (SS), secondary inorganic aerosols (SIA; ammonium, sulfate and nitrate) and organic and black carbons (OC, BC).**




| Aerosol types | Surface Concentration in µg/m$^3$ | Concentration at 1 km in µg/m$^3$ | MIAM, MICM, MINM in % | SSAM, SSCM in % | WASO in % | INSO in % | SOOT in % |
|---|---|---|---|---|---|---|---|
| Dust | 280 | 170 | 98 | - | 2 | - | - |
| DO. & DC. | 50 - 400 | - | >90 | - | - | - | - |
| Maritime Clean | 62 | 22 | - | 93 | 7 | - | - |
| MC. | 42 - 82 | 17 - 27 | - | 88 - 98 | - | - | - |
| Maritime Polluted | 70 | 25 | - | 83 | 16 | - | 1 |
| MPO. | 50 - 90 | - | - | 78 - 88 | - | - | - |
| MPC. | > 15 | - | - | 78 - 88 | - | - | - |
| Continental Clean | 9 | 8 | - | - | 59 | 41 | - |
| CC. | 1 - 20 | - | - | - | >50 | - | 0 - 1 |
| Continental Average | 25 | 22 | - | - | 58 | 40 | 2 |
| CA. | 9 - 35 | - | - | - | >50 | - | 1 - 3 |
| Continental Polluted | 50 | 45 | - | - | 66 | 30 | 4 |
| CP. | 20 - 70 | - | - | - | >50 | - | 3 - 6 |
| Urban | 105 | 93 | - | - | 56 | 36 | 8 |
| U. | 40 - 140 | - | - | - | >50 | - | 6 - 10 |

**Table 4 : Description of OPAC types (in grey); and selected conditions (in white) for selecting the MOCAGE profiles as input to the libRadTran radiative transfer model: concentrations of aerosols and proportion for each OPAC species Aerosol species of OPAC are mineral dust (nuc. mode: MINM; acc. mode: MIAM; coa. mode: MICM), sea salts (acc. mode: SSAM; coa. mode: SSCM), soluble aerosols (water soluble: WASO), insoluble aerosols (INSO) and soots (SOOT); the types used for the GSA are Desert Dust**
5 **over Ocean and Continent (DO. & DC.) Maritime Clean (MC), Maritime Polluted over Ocean and Continent (MPO & MPC), Continental Clean (CC), Continental Average (CA), Continental 5 Polluted (CP) and Urban (U).**


| Aerosol types | Surface Concentration in μg/m³ | Main species | Surface proportion over the total PM₁₀ |
|---|---|---|---|
| DO. & DC. | - | DD | |
| MC. | - | SS | SS > 85 % |
| MPO. | - | SS | SS < 85 % |
| MPC. | - | SS | SS < 85 % |
| CC. | 0 – 17 | IWS | |
| CA. | 17 – 34 | IWS | |
| CP. | 34 – 75 | IWS | |
| U. | > 75 | IWS | |

**Table 5 : Conditions on the NR profiles to be sorted into the in OPAC types before being used for assimilation. The first condition is the surface concentrations, the second is the main specie at the surface between Desert Dust (DD), Sea Salts (SS) and IWS (Insoluble, Water soluble, and Soot) and the third is a condition of the species over all the aerosols concentration. A species is described as a main species if its concentrations is above each other concentrations, for example DD is a main species if [DD]>[SS]**
5 **& [DD]>[IWS].**

| | Forecasts | SOA | Repartition of emissions from level 1 (surface layer) up to the 5th level |
|---|---|---|---|
| NR | ARPEGE | Yes | 52%; 26%; 13%; 6%; 3% |
| CR1 | IFS | Yes | 52%; 26%; 13%; 6%; 3% |
| CR2 | IFS | No | 52%; 26%; 13%; 6%; 3% |
| CR3 | IFS | No | 30%; 24%; 19%; 15%; 12% |
| CR4 | IFS | No | 100%; 0%; 0%; 0%; 0% |

**Table 6 : Table of differences between the NR simulation and the CRs simulations.**





| Hourly PM10 CRs « stations » vs NR « stations » | Bias (µg/m3) | RMSE (µg/m3) | FGE | FactOf2 | $R_P$ | $R_S$ |
|---|---|---|---|---|---|---|
| CR1 | -1.3 (-8.2 %) | 7.9 | 0.332 | 89.1 % | 0.671 | 0.748 |
| CR2 | -4.5 (-35.2 %) | 9.3 | 0.47 | 75.6 % | 0.609 | 0.709 |
| CR3 | -4.8 (-38.1 %) | 9.8 | 0.511 | 69.3 % | 0.572 | 0.671 |
| CR4 | -2.9 ( -20.5%) | 8.7 | 0.412 | 81.9 % | 0.623 | 0.712 |

**Table 7: Bias (B), RMSE, FGE, Factor of 2, Pearson correlation (Rp) and Spearman correlation (Rs) of the CRs simulation taking as reference the NR simulations for hourly PM$_{10}$ concentrations from January to April 2014. The comparison is made at the same station location as for AQeR stations.**

| Hourly PM2.5 CRs « stations » vs NR « stations » | Bias (µg/m3) | RMSE (µg/m3) | FGE | FactOf2 | $R_P$ | $R_S$ |
|---|---|---|---|---|---|---|
| CR1 | -0.8(-6.24%) | 5.9 | 0.307 | 91.1 % | 0.732 | 0.776 |
| CR2 | -3.9 (-37.4%) | 7.1 | 0.452 | 78.4 % | 0.69 | 0.731 |
| CR3 | -4.4 (-42.6%) | 7.6 | 0.505 | 70.6 % | 0.644 | 0.695 |
| CR4 | -1.8 (-15.1%) | 6.6 | 0.374 | 85.5 % | 0.665 | 0.73 |

**Table 8: Bias (B), RMSE, FGE, Factor of 2, Pearson correlation (Rp) and Spearman correlation (Rs) of the CRs simulation taking as reference the NR simulations for hourly PM$_{2.5}$ concentrations from January to April 2014. The comparison is made at the same station location as for AQeR stations.**



| Hourly PM10 CRs « stations » vs NR « stations » | Bias (µg/m3) | RMSE (µg/m3) | FGE | FactOf2 | $R_P$ | $R_S$ |
|---|---|---|---|---|---|---|
| AR1 | -1.17 (-7.21 %) | 7.16 | 0.296 | 92.2 % | 0.739 | 0.791 |
| AR2 | -2.91 (-21.3 %) | 8.1 | 0.373 | 85.3 % | 0.694 | 0.751 |
| AR3 | -3.53 (-26.2 %) | 8.67 | 0.417 | 80.4 % | 0.67 | 0.726 |
| AR4 | -0.756 (-5.31 %) | 8.03 | 0.339 | 88.2 % | 0.691 | 0.759 |

**Table 9: Bias, RMSE, FGE, Factor of 2, Pearson correlation and Spearman correlation of the ARs simulation taking as reference the NR simulations for hourly PM$_{10}$ concentrations from January to April 2014. The comparison is made at the same station location as for AQeR stations.**

| Hourly PM2.5 ARs « stations » vs NR « stations » | Bias (µg/m3) | RMSE (µg/m3) | FGE | FactOf2 | $R_P$ | $R_S$ |
|---|---|---|---|---|---|---|
| AR1 | -0.395 (-3.15%) | 5.61 | 0.284 | 92.7 % | 0.755 | 0.806 |
| AR2 | -2.28 (-20.5 %) | 6.31 | 0.364 | 86.6 % | 0.703 | 0.766 |
| AR3 | -2.94 (-27.1 %) | 6.86 | 0.416 | 80.9 % | 0.669 | 0.732 |
| AR4 | 0.109 (0.9 %) | 6.56 | 0.328 | 89.4 % | 0.699 | 0.765 |

5  **Table 10: Bias, RMSE, FGE, Factor of 2, Pearson correlation and Spearman correlation of the ARs simulation taking as reference the NR simulations for hourly PM$_{2.5}$ concentrations from January to April 2014. The comparison is made at the same station location as for AQeR stations.**



Figures :

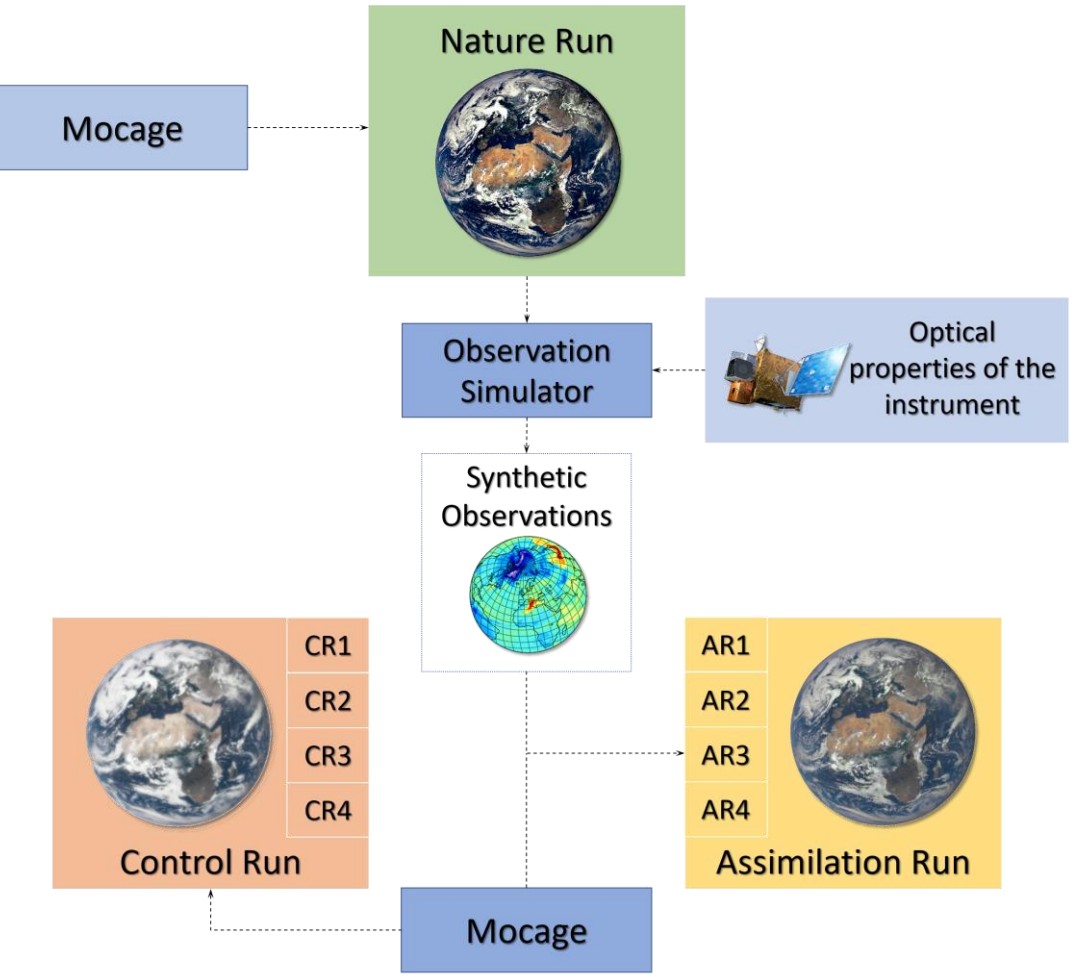

Figure 1: Schematic representation of the OSSE principle.



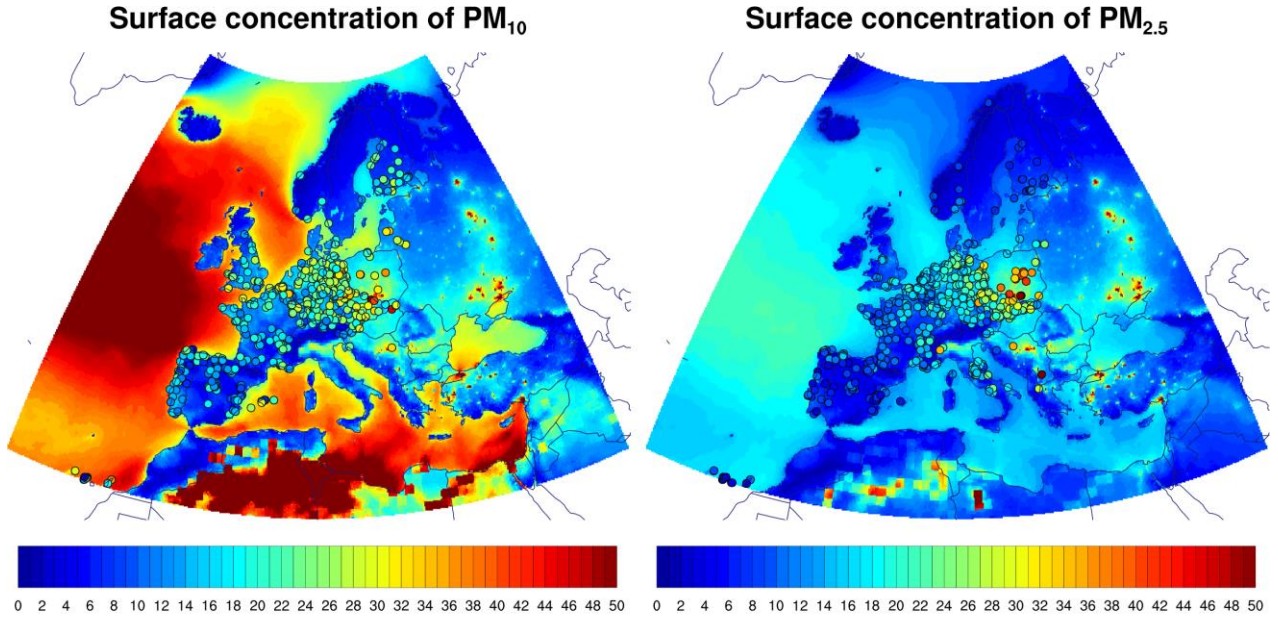

**Figure 2: Mean PM$_{10}$ (left panel) and PM$_{2.5}$ (right panel) surface concentration (µg.m$^{-3}$) of the NR (shadings) and AQeR station (color circles), from January to April 2014.**

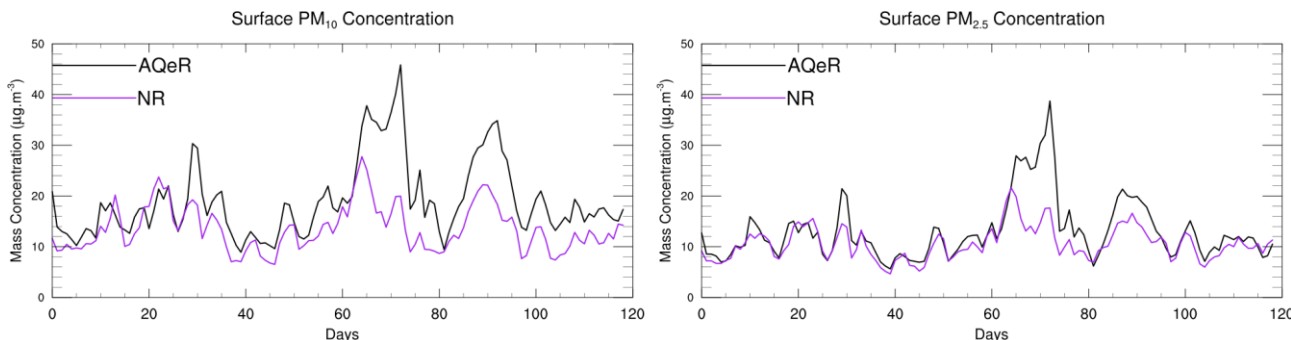

**Figure 3: Median surface concentration in µg.m$^{-3}$ of the NR (in purple) and the AQeR station (in black); the NR median concentration is calculated at the same location as for the AQeR stations. The left panel is for PM$_{10}$ surface concentrations while the right one is for PM$_{2.5}$.**







**Figure 4:** The FCI simulator. Step 1 is the computation of FCI radiance. Input parameters are the histograms of AOD, ozone total column, total water vapor content, ground albedo and solar zenithal angle. The libRadtran simulator simulates the distribution of radiance and reflectance in the VIS04 channel and takes into account the signal-to-noise ratio of FCI. Step 2 is the approximation of the reflectance in functions of key parameters using a Global Analysis Sensitivity method and Sobol indices. Step 3 is the retrieval of the AOD RMSE using random noise of measurement and the uncertainty of key parameters.





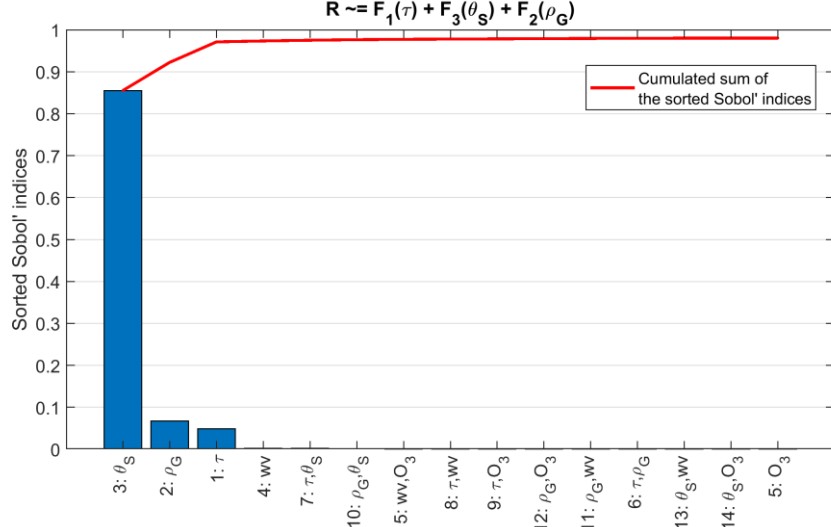

**Figure 5: Example of Sobol Indices for the output reflectance of VIS04, for the continental clean OPAC profile. In this case, the first three variables (the solar zenithal angle, the surface albedo and the AOD) explain more than 98% of the total variance of reflectance.**





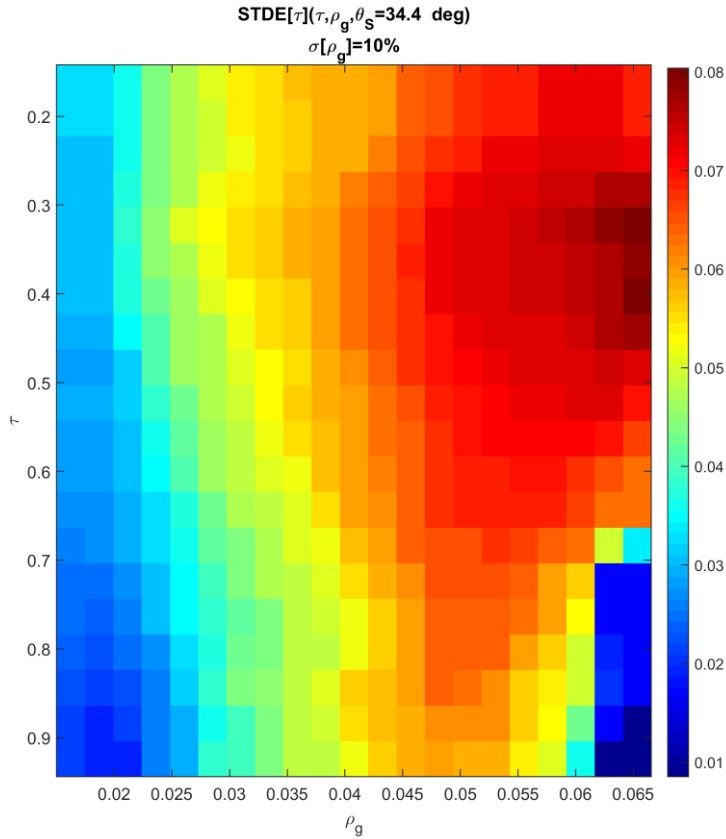

**Figure 6: RMSE of the AOD retrieval from the VIS04 reflectance, for the continental clean aerosol type and an uncertainty of 5 % on the ground albedo, and for a solar zenithal angle of 36 °, $\rho_g$ is the ground albedo and $\tau$ is the aerosol optical depth.**




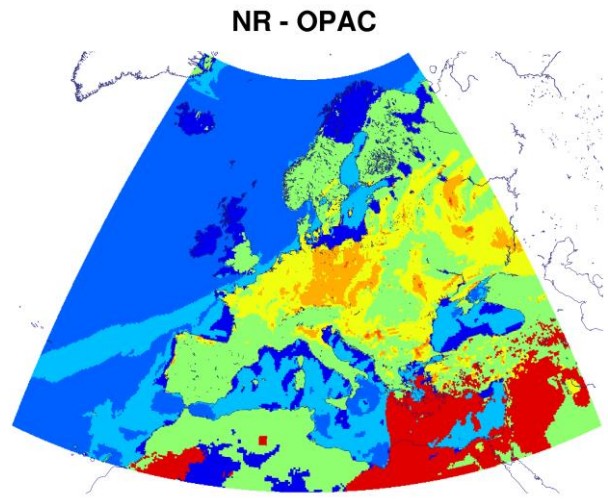

**Figure 7: Classification of the NR profiles for the 7ᵗʰ of March 2014 at 12 UTC. Deep Blue is for dismissed profiles, Blue is for Maritime Clean, Light Blue for Maritime Polluted, Green is for Continental Clean, Yellow is for Continental Average, Orange is for Continental Polluted, Deep Orange is for Urban, and Red is for Desert Dust.**

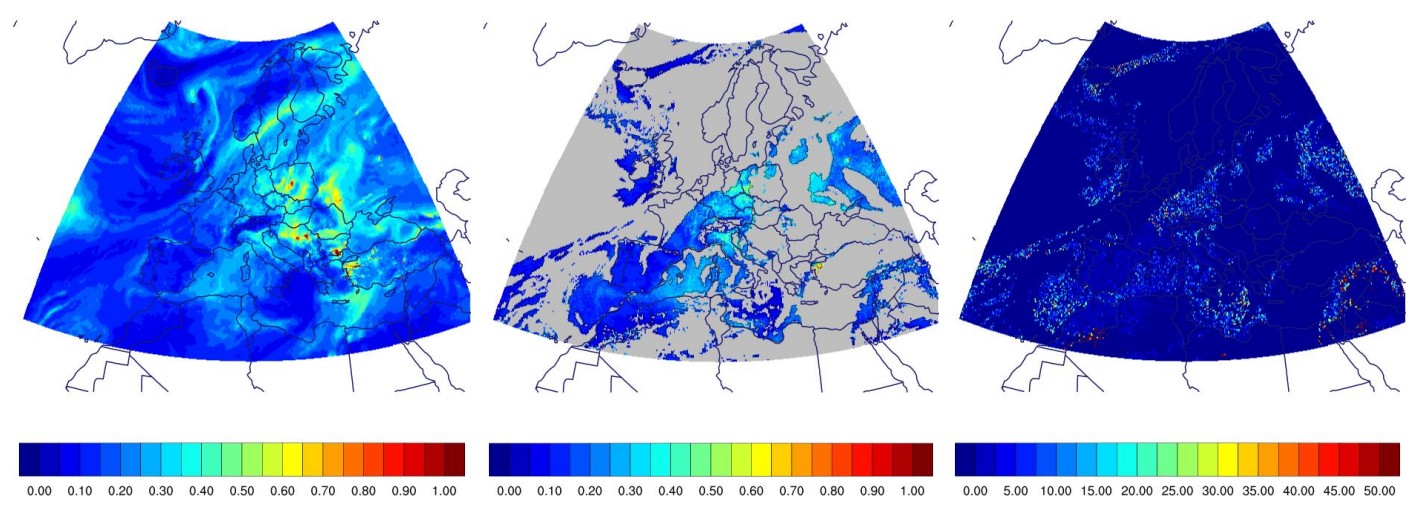

**Figure 8: NR's AOD the 7ᵗʰ of March 2014 at 12 UTC (left panel). Synthetic AOD (middle panel), the grey color represents the dismissed profiles. Relative differences between the NR and the synthetic observations (right panel).**



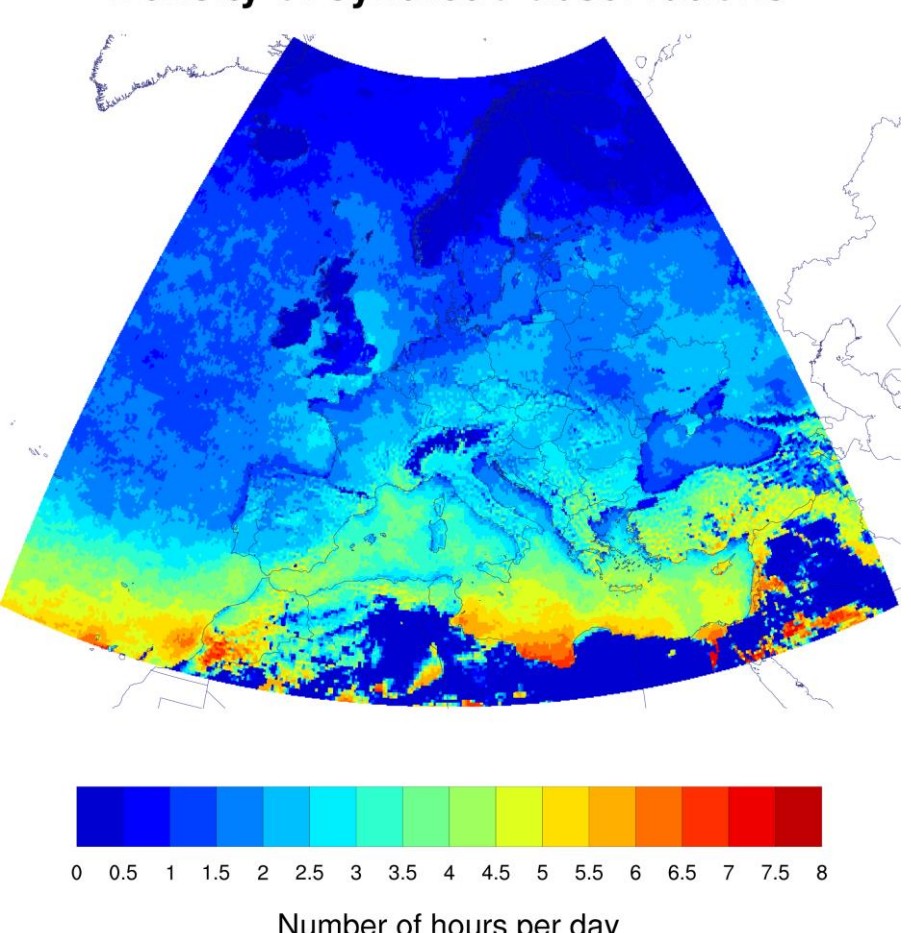

**Figure 9: Average (from January to April) number of selected profiles per day, avaible for assimilation.**



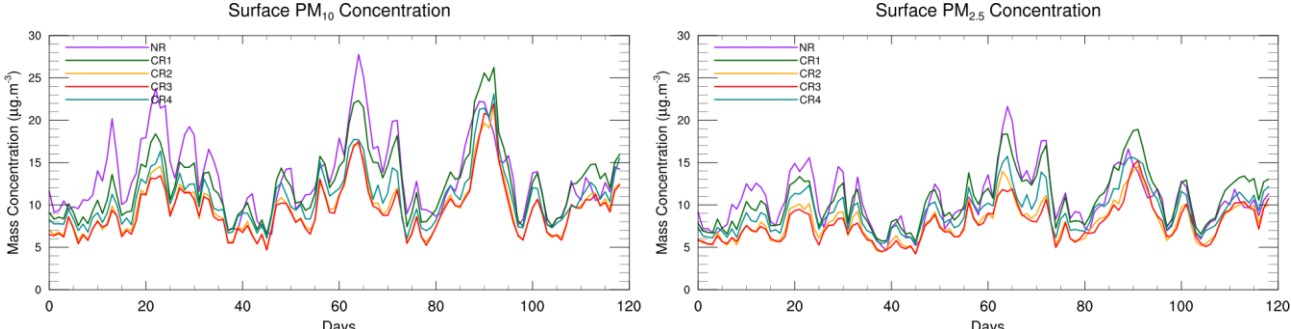

**Figure 10: Daily Median surface concentration of the NR (in purple) and the different CR (CR1 in green, CR2 in yellow, CR3 in red and CR4 in blue) determined for the same location as for the AQeR stations. The left graph is the PM$_{10}$ mass concentrations (µg.m$^{-3}$), while the right one represents the PM$_{2.5}$ mass concentrations.**

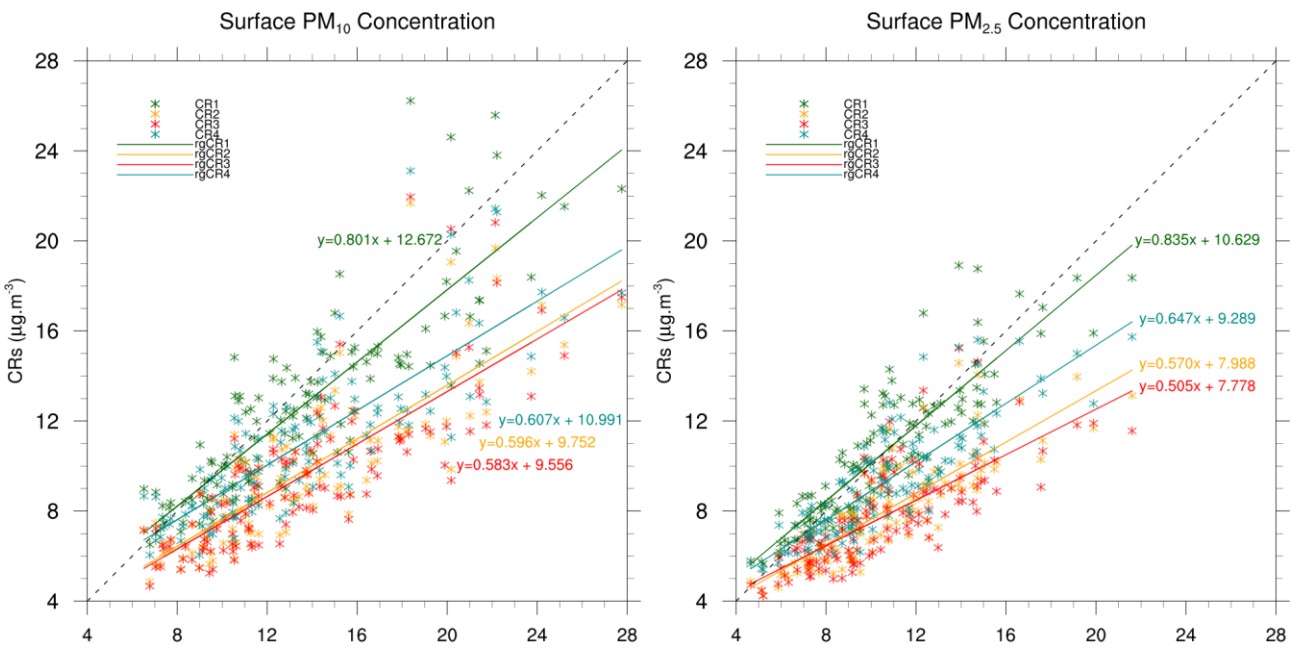

**Figure 11: Scatter plot of the CRs daily surface concentrations (µg.m$^{-3}$) as function of NR daily surface concentrations for PM$_{10}$ (left) and PM$_{2.5}$ (right), for virtual stations and from January to April 2014. rgCRX are the linear regressions of each dataset.**



**Figure 12: For each CR (CR1, CR2, CR3 and CR4), the figures represent a PM$_{10}$ comparison between the NR and the CRs from January to April 2014: the relative differences between CRs and the NR in %, the Pearson correlation and the fractional gross error.**





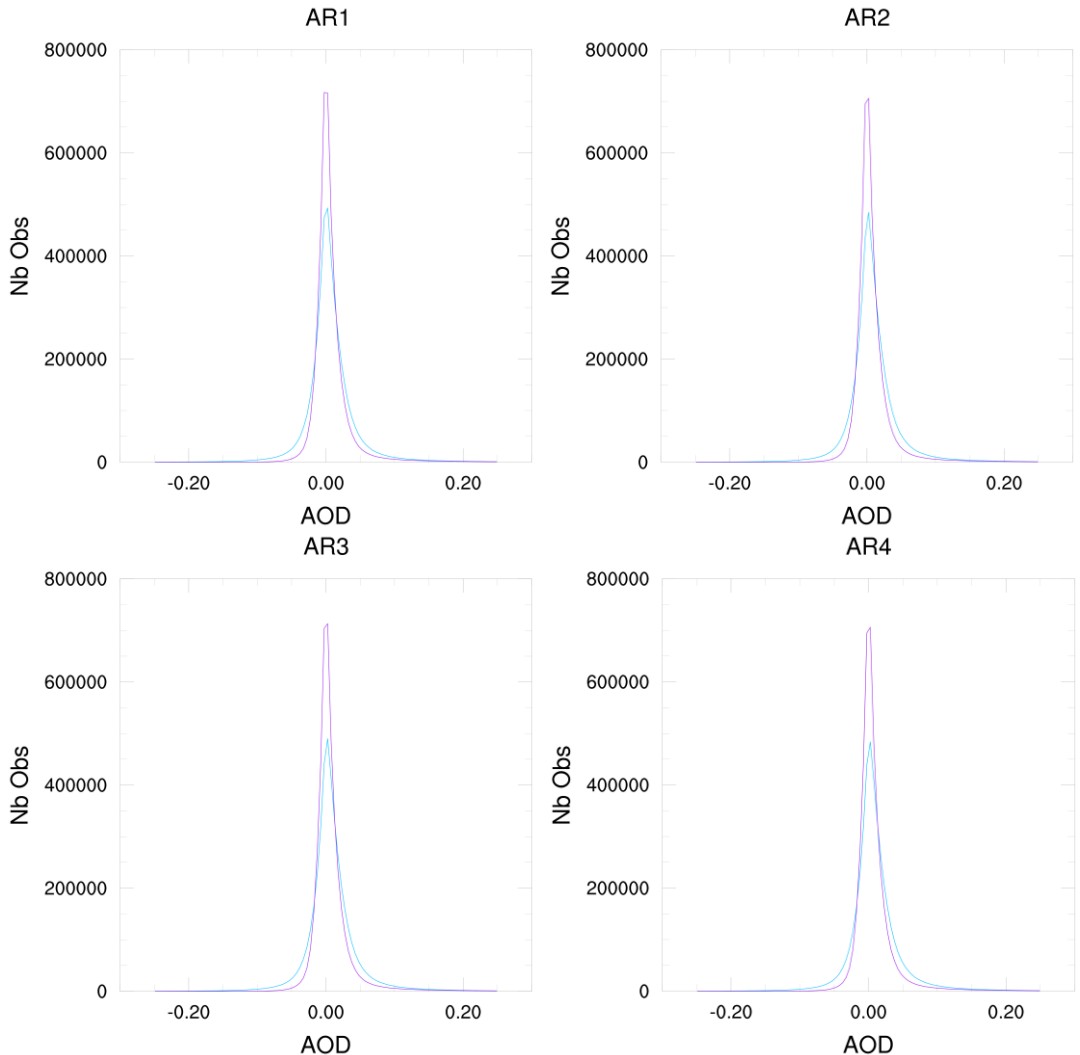

**Figure 13: Histograms of differences between synthetic observations and forecast fields (blue) and between synthetic observations and analyzed field (purple) for the four assimilation runs.**






**Figure 14 : Comparison of concentration in PM$_{10}$ between each AR (AR1, AR2, AR3 and AR4) and the NR from January to April 2014: the relative differences between ARs and the NR, the Pearson correlation and the fractional gross error.**





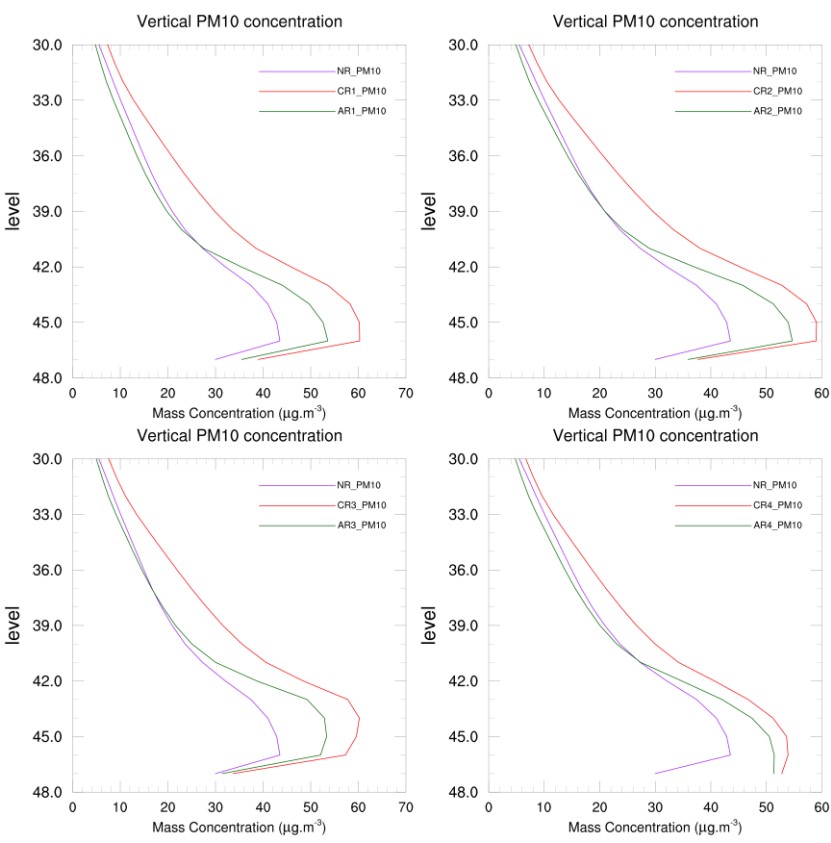

**Figure 15: Mean vertical profile, from January to April, over the domain of the concentrations (µg.m$^{-3}$) of PM$_{10}$ for the 4 set of simulations (1 in top left, 2 in top right, 3 in down left and 4 in down right). The NR is in purple, the CR is in red and the AR is in green.**





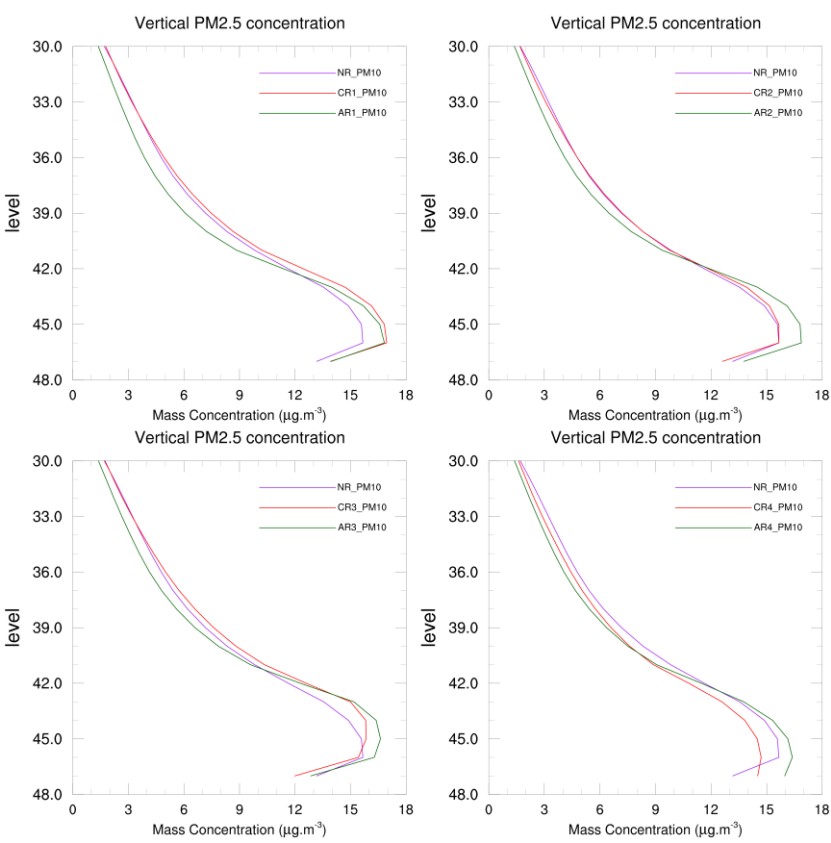

**Figure 16: Mean vertical profile, from January to April, over the domain of the concentrations (µg.m⁻³) of PM₂.₅ for the 4 set of simulations (1 in top left, 2 in top right, 3 in down left and 4 in down right). The NR is in purple, the CR is in red and the AR is in green.**





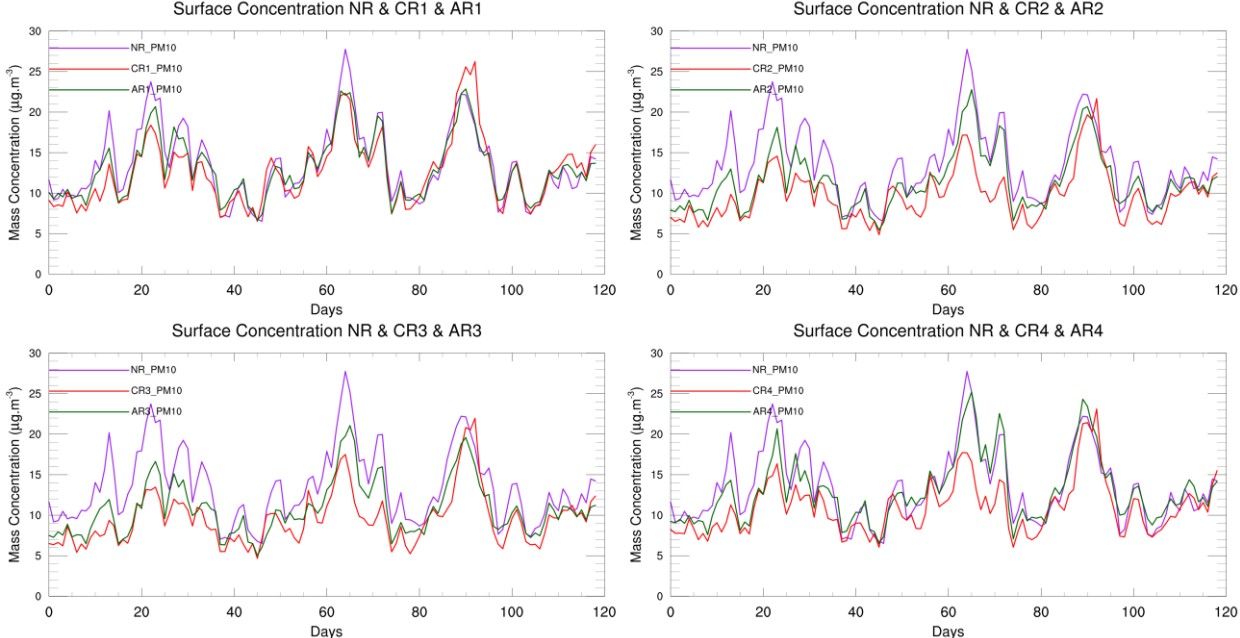

**Figure 17: Daily Median PM$_{10}$ surface concentration (µg.m$^{-3}$) of the NR (in purple) and the different CR (red) & AR (green) simulations (CR-AR-1 top left, CR-AR-2 top right, CR-AR-3 down left, CR-AR-4 down right) determined for the same location as for the AQeR station.**

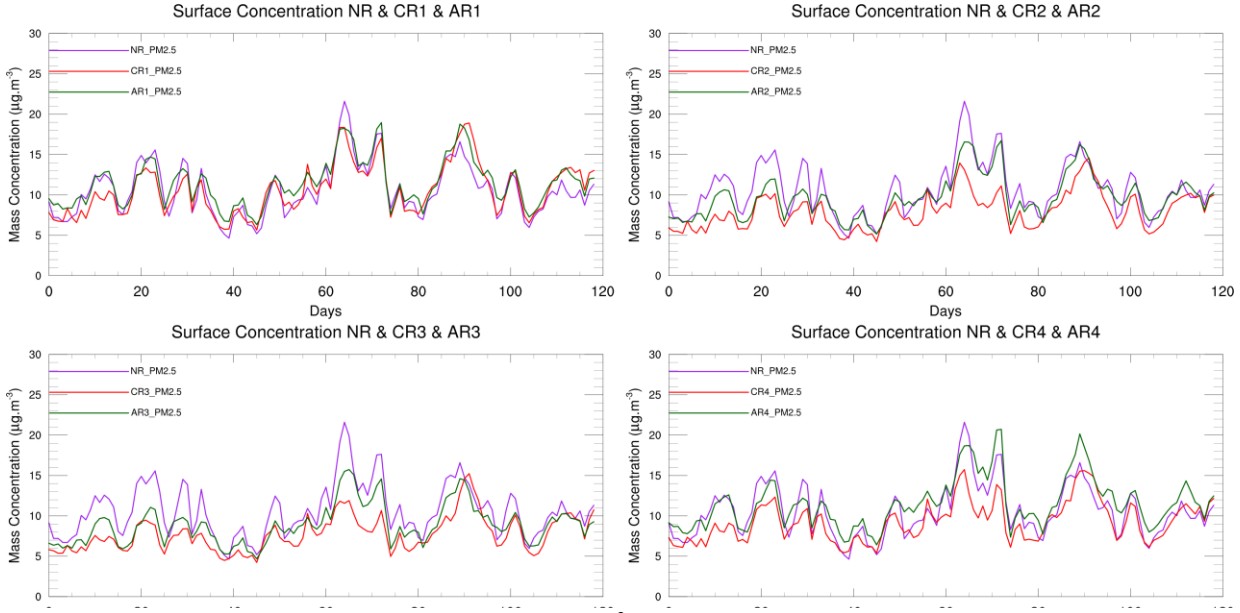

**Figure 18: Daily Median PM$_{2.5}$ surface concentration (µg.m$^{-3}$) of the NR (in purple) and the different CR (red) & AR (green) simulations (CR-AR-1 top left, CR-AR-2 top right, CR-AR-3 down left, CR-AR-4 down right) determined for the same location as for the AQeR station.**



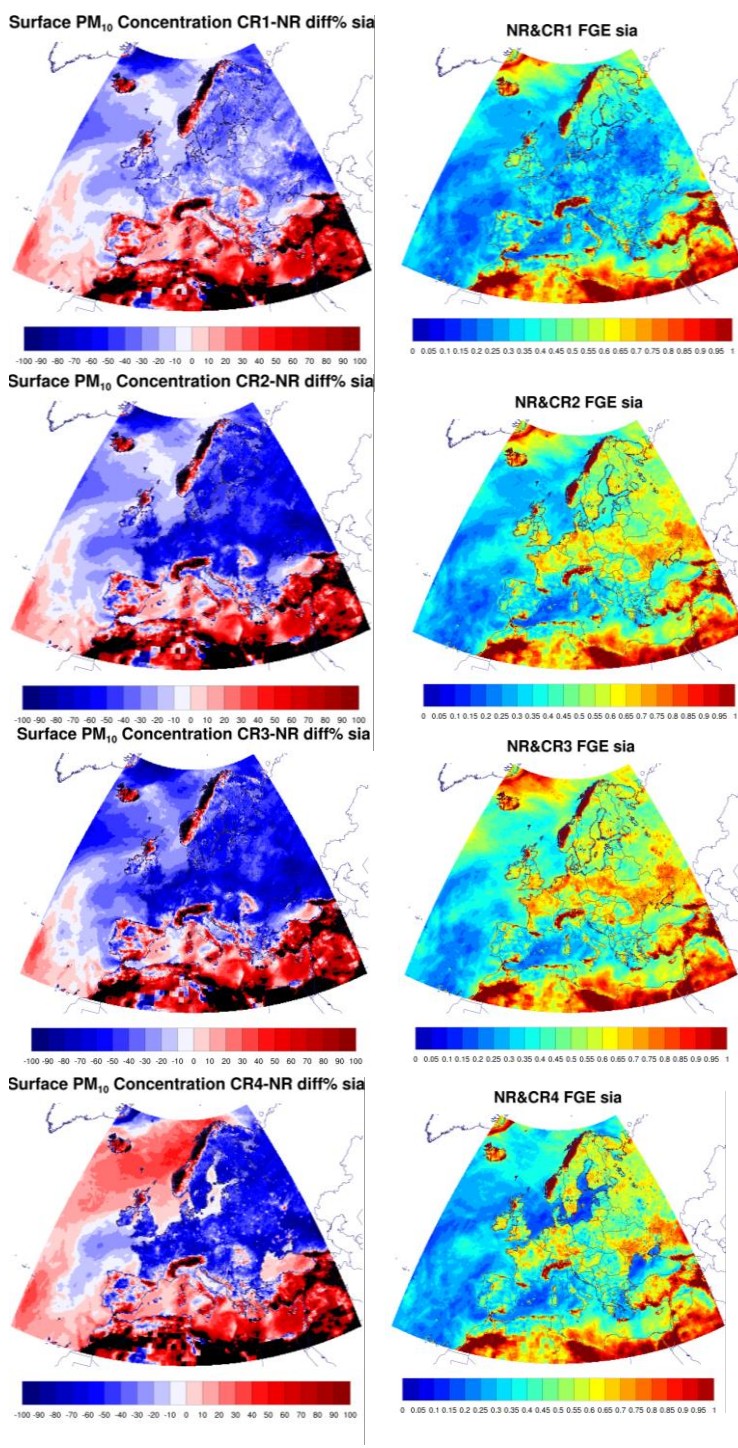

**Figure 19: PM$_{10}$ comparison between the NR and the CRs from the 7$^{th}$ March to the 15$^{th}$ March 2014: relative differences between CRs and the NR and fractional gross error.**





**Figure 20: PM$_{10}$ comparison between the NR and the ARs from the 7$^{th}$ March to the 15$^{th}$ March 2014: relative differences between ARs and the NR and fractional gross error.**