# Peer review of "Monitoring aerosols over Europe: an assessment of the potential benefit of assimilating the VIS04 measurements from the future MTG/FCI geostationary imager"

_Atmospheric Measurement Techniques, 2018_

## Referee Comment (RC1) · A. Benedetti (Referee) · 11 Oct 2018

**Review of "Monitoring aerosols over Europe: an assessment of the potential benefit of assimilating the VIS04 measurements from the future MTG/FCI geostationary imager" by Descheemaecker et al, 2018**

This paper discusses the assimilation of Aerosol Optical Depth (AOD) into the MOCAGE aerosol model using observations simulated from the future Flexible Combined Imager which will be flown on board the Meteosat Third Generation (MTG) satellite using a 3D-var approach. A Nature Run (NR) with the MOCAGE model is set-up to generate the "truth". The quality of this run for surface aerosol concentrations is verified using ground-based observations. The NR shows a low bias with respect to the ground-based measurements which is quite typical of global aerosol models. The NR is then used to generate the synthetic observations of AOD to be then assimilated. To ensure the robustness of OSSE results, 4 control runs (CRs) are also set-up. While the underlying model is the same as in the NR, the CRs different in either initialization or in the aerosol parameterizations/localizations of the emissions. This is to obviate the problem of using the same model to perform the NR and the subsequent assimilation and have at least a (small) ensemble of runs to judge the quality of the assimilation. The experiments with assimilation of the synthetic measurements (Assimilation Runs, ARs) are closer to the NR than the CRs, indicating a successful assimilation. The ARs are also compared with ground-based particulate matter (PM) observations. Results show that the assimilation of the simulated AOD reduces the error with respect to independent surface-based observations, even if the underestimation is still present.

The paper is interesting and illustrates an important application of geostationary satellite retrievals for aerosol assimilation which should become more prominent in the future as more geostationary satellites around the world will carry advanced imagers. These datasets will complement those from polar-orbiting satellites to provide high-resolution and high-frequency data on the state of the atmospheric particulate fields.

I have a few questions/comments, which should be addressed before the manuscript can be published.

Major comments:

1. The procedure to create the synthetic datasets it not clear. What is clear is that a lot of effort and time has gone into this part of the work, but then the procedure is introduced in what it looked to me a chaotic way. First you talk about the Monte Carlo approach to calculate the radiances, then you mention a global sensitivity analysis whose purpose is not clear, then you mention the use of the OPAC tables. I was not sure whether the AODs were retrieved from the simulated FCI radiances and how. I really got lost in this section. Please rewrite it all keeping in mind that most readers are not familiar with how to create a synthetic AOD dataset. Give enough details, but do not get carried away.

2. At the end of section 3, you conclude that only a few profiles can be assimilated over the continent. This is an important result but at the same time undermines the concept of the paper which is to highlight the usefulness of the high temporal resolution data from the geostationary sensors. Besides, it is important to keep the study realistic, but with an OSSE you can go wild and seek to demonstrate the untapped potential of the instrument. For example, what would happen if you were able to use the full temporal resolution of the

instrument and not only hourly data? Have you thought of these issues? What is completely unrealistic and what is pushing the limit of the technology? I do not feel you had the time to address these important questions. While this may be subject of future work, you need to comment on this and add your insight.

3. You only selected one wavelength (444nm). I am sure this was due to the amount of work needed to generate the synthetic dataset but your choice needs to be better justified as it is rather limiting.

4. In some parts the paper reads too academic. For example, the long list of verification metrics including the formulas is not needed. Please change that.

5. For all figures, particularly the PM maps, the legends have to be bigger.

6. Some of the tables can be eliminated. I found table 3 particularly cumbersome. Please consider presenting the information in a more concise way.

Other comments:

Line 20: please use also this reference. Peuch, V. and Engelen, R.: Towards and operational GMES Atmosphere Monitoring Service, ECMWF Newsletter, 132, 20–25, 2012.

Line 20: Please expand the overview of the OSSE approach and provide more references.

Line 10: are 6 bins used for all aerosols, including sulphates, nitrates etc?

Line 23: "parameterized" instead of "made"

Line 9: Please add Benedetti et al, 2009 (Benedetti, A., Morcrette, J.-J., Boucher, O., Dethof, A., Engelen, R., Fisher, M., Flentje, H., Huneeus, N., Jones, L., Kaiser, J., et al.: Aerosolanalysis and forecast in the European centre for medium-range weather forecasts integrated forecast system: 2. Data assimilation, Journal of Geophysical Research: Atmospheres (1984–2012), 114, 2009) to the reference list for the AOD assimilation approach.

Line 18. Is the computation of the optical properties performed online or off-line and tabulated in a look-up table.

Lines 10-30. Please rewrite in a less academic way. Maybe you do not need to put formulas for all of the metrics.

Page 8.

Line 7. This is not useful, please explain what 1-5 means.

Line 12: remove "a bit". The underestimation of PM in global aerosol models is a general problem, due to unresolved emissions and coarse resolution. Even a resolution of 0.2 does not allow to resolve all the regional and urban pollution features.

Line 30. It is unusual that AOD is overestimated. The explanation on page 9 line 10 is unlikely.

Line 23: why was only one wavelength selected, and why the 444nm?

Referring to the general comments this section is not clear to me and would benefit from substantial rewriting.

Lines 24-32. You are effectively saying that your experimental set-up is not adequate to explore the potentiality of the instrument due to limited spatial resolution. This is actually not ideal for an OSSE. Would it be possible to run the model at higher resolution or downscale somehow the synthetic AOD? I am not suggesting this extension for this study, but perhaps for a follow-on.

Line 5. Please explain how convergence is connected to thinning. Usually thinning is applied to avoid using correlated observations in the assimilation, without accounting for the correlation errors (off-diagonal elements of the R matrix).

Line 18. Use another adjective instead of "great".

Line 19. I missed the supplementary material. Was it accessible?

Line 8-9. Where does the improvement in the vertical profile come from? AOD does not contain information in the vertical distribution of the aerosols.

Section 6.

The conclusions are fine but this is where you should elaborate more what you would do to extend this study. For example you could comment on trying to increase the resolution of the model to produce synthetic measurements that are closer to the future capabilities of FCI (or something along these lines).

---

## Referee Comment (RC2) · R. Timmermans (Referee) · 15 Oct 2018

**General comments**

This paper presents the results from an OSSE study to assess the impact of the FCI AOD observations from a geostationary instrument on continental scale PM10 analyses. The study is significant as there is a lot of attention for new instruments on geostationary instruments and their usefulness in air quality applications. OSSEs are useful tools to evaluate the value of the new instruments prior to their operation or even prior to their production. In the study the authors have put large effort in the realism of the OSSE and demonstrating this, which is positive and provides trust in the results.

The paper is well structured and leads you through the different steps of the experiment.

The main comments I have are related to the choices made in the study and the amount of discussion.

- To identify the value of the new instrument it should ideally be compared to the current situation. Currently most operational models assimilate surface observations of PM10 and PM2.5 and some also AOD observations from e.g. MODIS. It would be beneficial to show the added value of the FCI observations on top of these common observations. How beneficial are these observations from the geostationary instrument as compared to observations from an orbiting satellite with lower temporal resolution. Perhaps an experiment could be done with assimilation of only one observation per day, as compared to the hourly observations, do you see an added value? Can the satellite observations replace or add value to the surface observations (which are much cheaper). This is something is really miss in the discussion to support the value of this instrument for aerosol modelling.

- I would suggest more room for discussion of the results. There is a large focus on the areas/cases where it goes well, but some discussion is required on the situations where it does not work so well. The assimilation distributes the increments based on the fractions, so it can not correct for errors in the size distributions. The same holds for the vertical distributions. The AOD does not provide any vertical information, so what happens when the vertical profile is wrong, such as for example in figure 16 bottom right, you can see that the system does not work in these cases. So when the AOD in underestimated but the surface PM is overestimated (or the other way around), the AOD observations will increase the PM at all levels also at the surface, leading to worse results. This discussion on AOD-PM relationship and the importance of having correct vertical distributions or vertical information is missing and should be added. Especially since you are showing that your NR shows overestimation of AOD and underestimation of PM, when you would use that version of the model for assimilation I think you would get problems.

**Specific comments**

- The model runs are performed at a resolution of approximately 20km which is soon not really representative anymore for regional air quality forecasts/analysis. Many models already run at a 0.1 deg resolution. Especially since the observations are available at a 2km resolution I wonder why it is not chosen to run at a higher resolution. Somewhere it is mentioned that the goal of the study is to evaluate the impact for continental modelling, Please elaborate a bit more on this choice and for the

discussion part I would add the potential to look at the impact at urban scales, maybe in a follow-up study.

- For the vertical distribution of the aerosol emissions, a fixed profile is used as I understand correctly, is this not depending on the source type of the emissions, e.g. car emissions at the surface and emissions from industry higher up?

- The inclusion or exclusion of SOA is not clear throughout the paper. Somewhere it is mentioned that SOA is added to the NR and CR1 by using a percentage of the primary carbon species. But further along it looks as if the SOA is not included in the computation of AOD (p9, line 16-17), is that correct? Your nature run is the "real world' so then also the synthetic observations should include the SOA. What is the impact of excluding SOA in the AOD synthetic observations?

- Section 3 is very hard for me to follow, it is not within my field of expertise but I get the impression that the AOD is only computed from the model concentrations without taking into account any radiate transfer modelling? Is this a correct assumption from my side? because then the sensitivity of the instrument to different altitudes in the atmosphere is not taken into account which can lead to overoptimistic results. On the other hand the errors are computed in a very accurate way. A lot of attention goes to the simulation of these errors, which are very important for the realism of the OSSE. However I think the amount of figures/tables and text dedicated to this part of the study is out of proportion and needs condensation and rewriting.

- p 11, Filtering: A lot of observations are removed due to filtering. This is an important comment. The added value of geostationary satellites lies in their temporal resolution. If you only retain 1 to 4 observations per day, is there still a large added value, is this representative of the future real situation? Please add some discussion

- Location of observations, the observations seem to be concentrated over central Europe, how representative are the results at the AQeR for other regions? You can see that the plots for the entire domain provide different conclusions than the plots for the AQeR stations.

- Validity of CR-NR: The statistical metrics have been compared to metrics from literature. Two different papers have been used, but if I am correct both evaluating the ensemble of models which is always better than the individual models. Have you also compared to individual model results. The CR-NR seem to be smaller than the NR − real observations difference.

- Only one observation per hour is used because of the system.  As is suggested in the conclusion I would make a super observation. It is mentioned that avoiding overoptimistic results is one of the reasons for this choice but I do not agree as this will probably be done once the observations become available in real life, so I do not see why this would lead to overoptimistic results

- Spatially 3 out of 4 observations are removed trough filtering for convergence of cost function, while already a lot of observations are removed due to clouds etc. Is a larger impact of the observations foreseen without this spatial filtering?

**Technical corrections**
- p1., line 17 Abstract, change 4-month**s** to 4-month

- p2. Line 6. which lead → which lead**s**

- p2, line 13, Only the WHO limit values are mentioned, but it would also be good to include the official EU limit values.

- p2, line 20, there are more appropriate references for the CAMS services, please add the website and the paper from Marécal which is use further along in the paper.

- p2 line 30, MODIS is now also available in a 1x1 km product (MAIAC)

- p4, line 10-11 this sentence is unclear to me, do you mean by combining AOD and error characteristics?

- p4, line 12, CR, which should represent……(something like the current situation, the situation without use of the observations)

- p4, line 25-26. Also when you are using two different models, you should evaluate this.

- p4, line 27, → as MOCAGE is used for **both**….

- p7, line 20, FGE is **also** used

- Figure 2, I find it very hard to see the NR background in central Europe with all the overlying circles and the small plots. It is mentioned that the variability and maxima are well represented, but I cannot evaluate this when I do not see the background.

- p8, lines 20-24. The underestimation is indeed common, I do think there are many more possible reasons for this, such as underestimation of emissions in cold winter periods, and perhaps the modelling of stable winter conditions with shallow surface layers.

- P8, line 16: maxima, I would change this word, as I relate maxima to the absolute maximum values, while I think you mean the location of the mamixum values.

- p12, line 5 slowest → slower

- p13, line 2, especially the CR4: but the bias for CR4 is quite small….

- p13, line 24, here the purpose of the paper is mentioned, but this should be stated more clearly in the introduction, especially the focus on the continental scale.

- p14, line 25-28 Figure 14 versus figure 12, I found it hard to see the improvement, while a large improvement is mentioned, maybe it would be helpful to direct the reader to some specific areas where it is visible. Tables 9 and 10 are clear but only cover central Europe.

- p15, lines 5-10 please add here the discussion of AOD-PM relation as suggested in the general comments

- p15, line 29-20, what is meant with high spatial and temporal episode?

- p16, summary, please also add the case where it does not work (simulation 4 at the surface, averaged over whole domain).

---

## Author Comment (AC1) · 2 Jan 2019

Dear Editor, Dear reviewers,

We thank both reviewers for their positive and constructive comments on the first version of the manuscript. We have taken into account their suggestions as described below. These corrections have led to a new version of the manuscript, which we hope to be considered for final publication in *Atmospheric Measurement Techniques*.

In this letter, we provide response to all Reviewer Comments. For a clear and easy-to-follow sequence, every comment is copied in Italic and is followed by our answer, in which the changes that have been done in the new version of the manuscript are addressed. The page and line references can be found in the "track-change" file.

**Response to Reviewer Comments 1**

*Major comments*

1. *The procedure to create the synthetic datasets it not clear. What is clear is that a lot of effort and time has gone into this part of the work, but then the procedure is introduced in what it looked to me a chaotic way. First you talk about the Monte Carlo approach to calculate the radiances, then you mention a global sensitivity analysis whose purpose is not clear, then you mention the use of the OPAC tables. I was not sure whether the AODs were retrieved from the simulated FCI radiances and how. I really got lost in this section. Please rewrite it all keeping in mind that most readers are not familiar with how to create a synthetic AOD dataset. Give enough details, but do not get carried away.*

This section has been completely re-written, aiming at clarifying the methodology and at pointing out the essential steps and results. As you may see in the new version of the manuscript (Section 3), the method consists in adding an error perturbation to the NR AOD, which variance is deduced from a sensitivity analysis taking into account the FCI characteristics. The sensitivity analysis is used to compute a look-up-table that provides the RMSE of AOD as a function of key parameters. This methodology has never been published and so we consider that it deserves to be presented and explained with enough details. But we acknowledge that some unnecessary details were misleading in the previous version of the manuscript. They have been removed (two tables and two figures) or summarized. Also, part of the method description, that is not essential to the understanding of the article, have been exported in an appendix.

2. *At the end of section 3, you conclude that only a few profiles can be assimilated over the continent. This is an important result but at the same time undermines the concept of the paper which is to highlight the usefulness of the high temporal resolution data from the geostationary sensors. Besides, it is important to keep the study realistic, but with an OSSE you can go wild and seek to demonstrate the untapped potential of the instrument. For example, what would happen if you were able to use the full temporal resolution of the instrument and not only hourly data? Have you thought of these issues? What is completely unrealistic and what is pushing the limit of the technology? I do not feel you had the time to address these important questions. While this may be subject of future work, you need to comment on this and add your insight.*

Since we consider assimilation of FCI data for regional-scale pollution with a 0.2° model, the study does not fully exploit the temporal and horizontal high-resolution of FCI. Our objective is to assess how much the VIS04 channel is useful for air pollution in Europe with a 0.2° resolution model using hourly observations. We agree that out study does not fully exploit the potential of FCI, and that this question needs to be addressed address. To clarify this question, the new version of the manuscript :

- In the introduction, it has been clarified that the study focuses on regional scale pollution (page 3, line 29),
- Includes a discussion (Section 6), in which a whole paragraph addresses the potential of the high-resolution of FCI,
- In the conclusion, a perspective has been added about the exploitation of FCI at high resolution (page 17, lines 26-29).

3. *You only selected one wavelength (444nm). I am sure this was due to the amount of work needed to generate the synthetic dataset but your choice needs to be better justified as it is rather limiting.*

In the introduction, the choice of this wavelength has been justified by the fact that it is new compared to SEVIRI and that it is the shortest one, theoretically favourable for the detecting fine particles. In the new manuscript, in order to clarify and expand the justification:

- because MOCAGE cannot assimilate AOD at several wavelengths simultaneously, only one wavelength was chosen. The introduction brings forward the argument (page 3, line 31) that the study focuses on the more relevant wavelength (the shortest) *a priori for fine aerosol detection*,
- In Section 2.3 (*Assimilation system PALM*), we provide more argument why MOCAGE cannot assimilate AOD from several wavelengths simultaneously; a sentence has also been added at the beginning of Section 3.

4. *In some parts the paper reads too academic. For example, the long list of verification metrics including the formulas is not needed. Please change that.*

The formulas and definition of verification metrics have been removed (page 7).

5. *For all figures, particularly the PM maps, the legends have to be bigger.*

The PM maps have been zoomed in and their legends have been improved, in a similar manner for the figures of the article and of the supplementary material document.

6. *Some of the tables can be eliminated. I found table 3 particularly cumbersome. Please consider presenting the information in a more concise way.*

Table 3 has been removed and reference to Ceamanos et al (2014), where the same information may be found, has been added (page 18, line 10). Some other figures and tables, mostly related to Section 3, could be removed from the manuscript without losing the essential information, so they have been removed.

*Other comments:*

*Page 2 Line 20: please use also this reference. Peuch, V. and Engelen, R.: Towards and operational GMES Atmosphere Monitoring Service, ECMWF Newsletter, 132, 20–25, 2012.*
Done (page 2, line 25)

*Page 3 Line 20: Please expand the overview of the OSSE approach and provide more references.*
The end of the introduction (page 4, lines 1-11) has been re-ordered to make a specific paragraph on the OSSE approach. This paragraph has been expanded with more references and now it points out the main potentialities and limitations of OSSEs

*Page 5 Line 10: are 6 bins used for all aerosols, including sulphates, nitrates etc?*
Done (page 5, line 20)

*Page 5 Line 23: "parameterized" instead of "made"*
Done (page 6, line 1)

*Page 6 Line 9: Please add Benedetti et al, 2009 (Benedetti, A., Morcrette, J.-J., Boucher, O., Dethof, A., Engelen, R., Fisher, M., Flentje, H., Huneeus, N., Jones, L., Kaiser, J., et al.: Aerosol analysis and forecast in the European centre for medium-range weather forecasts integrated forecast system: 2. Data assimilation, Journal of Geophysical Research: Atmospheres (1984–2012), 114, 2009) to the reference list for the AOD assimilation approach.*
Done (page 3 line 20, and page 6 line 17)

*Page 6 Line 18. Is the computation of the optical properties performed online or off-line and tabulated in a look-up table.*
It is done off-line ; we have completed the manuscript accordingly (page 7 lines 26-27)

*Page 7 Lines 10-30. Please rewrite in a less academic way. Maybe you do not need to put formulas for all of the metrics.*
Done

*Page 8. Line 7. This is not useful, please explain what 1-5 means.*
Done (page 7, lines 29-30)

*Page 8 Line 12: remove "a bit". The underestimation of PM in global aerosol models is a general problem, due to unresolved emissions and coarse resolution. Even a resolution of 0.2 does not allow to resolve all the regional and urban pollution features.*
Done (page 8, line 1)

*Page 8 Line 30. It is unusual that AOD is overestimated. The explanation on page 9 line 10 is unlikely.*
At the global scale, Sic et al (2015) showed also that MOCAGE tends to overestimate AOD, despite PM at surface are underestimated. We aknowledge that other models tend also to underestimate AOD (Morcrette et al, 2009). The comparison of our NR with AERONET stations show that the bias is almost positive for all stations in Europe. This can be explained by assumptions in MOCAGE and in the AOD calculations : the vertical profiles of emission injections, the size distribution of aerosols and the hypotheses that underlie the computation of optical properties.
This part of the manuscript has been rephrased (page 8, lines 24-34).

*Page 9 Line 23: why was only one wavelength selected, and why the 444nm?*
*Referring to the general comments this section is not clear to me and would benefit from substantial rewriting.*

Done

*Page 13 Lines 24-32. You are effectively saying that your experimental set-up is not adequate to explore the potentiality of the instrument due to limited spatial resolution. This is actually not ideal for an OSSE. Would it be possible to run the model at higher resolution or downscale somehow the synthetic AOD? I am not suggesting this extension for this study, but perhaps for a follow-on.*
A discussion (Section 6) has been added, that addresses this question.

*Page 14 Line 5. Please explain how convergence is connected to thinning. Usually thinning is applied to avoid using correlated observations in the assimilation, without accounting for the correlation errors (off-diagonal elements of the R matrix).*
Our argument that thinning is useful to speed up convergence is based on some experiments that we have done using these observations and MOCAGE-PALM. We have rephrased this sentence (page 12, lines 27-28).

*Page 14 Line 18. Use another adjective instead of "great".*
« great » has been removed (page 13, line 10).

*Page 14 Line 19. I missed the supplementary material. Was it accessible?*
Yes, it is on the same webpage as the article, besides the manuscript.

*Page 16 Line 8-9. Where does the improvement in the vertical profile come from? AOD does not contain information in the vertical distribution of the aerosols.*
We have added an argument there (page 15, lines 2-4), based on the published work of Sic et al (2016).

*Section 6. The conclusions are fine but this is where you should elaborate more what you would do to extend this study. For example you could comment on trying to increase the resolution of the model to produce synthetic measurements that are closer to the future capabilities of FCI (or something along these lines).*
This question is addressed in the discussion (Section 6). Besides, the conclusion has also been expanded following this line.

**Response to Reviewer Comments 2**

*- To identify the value of the new instrument it should ideally be compared to the current situation. Currently most operational models assimilate surface observations of PM10 and PM2.5 and some also AOD observations from e.g. MODIS. It would be beneficial to show the added value of the FCI observations on top of these common observations. How beneficial are these observations from the geostationary instrument as compared to observations from an orbiting satellite with lower temporal resolution. Perhaps an experiment could be done with assimilation of only one observation per day, as compared to the hourly observations, do you see an added value? Can the satellite observations replace or add value to the surface observations (which are much cheaper). This is something is really miss in the discussion to support the value of this instrument for aerosol modelling.*
In the new discussion section, we have added a discussion on this issue. This discussion includes the results of a new simulation called AR3LEO, in which only pixels at 12UTC are assimilated every day.

*- I would suggest more room for discussion of the results. There is a large focus on the areas/cases where it goes well, but some discussion is required on the situations where it does not work so well. The assimilation distributes the increments based on the fractions, so it can not correct for errors in the size distributions. The same holds for the vertical distributions. The AOD does not provide any*

*vertical information, so what happens when the vertical profile is wrong, such as for example in figure 16 bottom right, you can see that the system does not work in these cases. So when the AOD in underestimated but the surface PM is overestimated (or the other way around), the AOD observations will increase the PM at all levels also at the surface, leading to worse results. This discussion on AOD-PM relationship and the importance of having correct vertical distributions or vertical information is missing and should be added. Especially since you are showing that your NR shows overestimation of AOD and underestimation of PM, when you would use that version of the model for assimilation I think you would get problems*

In section 6, some more discussion on how important the vertical distribution of aerosols is and about the relationship between AOD bias and PM bias has been added. This discussion follows the presentation of the results in Section 5 about the vertical distribution of PM of CR3 and CR4 and the biases at surface of AR4.

*- The model runs are performed at a resolution of approximately 20km which is soon not really representative anymore for regional air quality forecasts/analysis. Many models already run at a 0.1 deg resolution. Especially since the observations are available at a 2km resolution I wonder why it is not chosen to run at a higher resolution. Somewhere it is mentioned that the goal of the study is to evaluate the impact for continental modelling, Please elaborate a bit more on this choice and for thediscussion part I would add the potential to look at the impact at urban scales, maybe in a follow-up study.*

Discussion has been added on this issue (Section 6).

*- For the vertical distribution of the aerosol emissions, a fixed profile is used as I understand correctly, is this not depending on the source type of the emissions, e.g. car emissions at the surface and emissions from industry higher up*

Yes, a fixed profile is used for all source type. This is the usual MOCAGE setup.

*- The inclusion or exclusion of SOA is not clear throughout the paper. Somewhere it is mentioned that SOA is added to the NR and CR1 by using a percentage of the primary carbon species. But further along it looks as if the SOA is not included in the computation of AOD (p9, line 16-17), is that correct? Your nature run is the "real world' so then also the synthetic observations should include the SOA. What is the impact of excluding SOA in the AOD synthetic observations*

SOA are included in the NR (page 7, lines 19-20) and CR1 (page 10, line 22) and not in other CR (consistently with Table 4). SOA are included in the AOD synthetic observations (like all types of aerosols, as stated in Section 3).

*- Section 3 is very hard for me to follow, it is not within my field of expertise but I get the impression that the AOD is only computed from the model concentrations without taking into account any radiate transfer modelling? Is this a correct assumption from my side? because then the sensitivity of the instrument to different altitudes in the atmosphere is not taken into account which can lead to overoptimistic results. On the other hand the errors are computed in a very accurate way. A lot of attention goes to the simulation of these errors, which are very important for the realism of the OSSE. However I think the amount of figures/tables and text dedicated to this part of the study is out of proportion and needs condensation and rewriting.*

This section has been completely re-written, aiming at clarifying the methodology and at pointing out the essential steps and results. A radiative transfer model (LibRadTran, see the Appendix) is indeed taken into account for the computation of errors. The methodology to compute AOD errors has never been published and so we consider that it deserves to be presented and explained with

enough details. But some unnecessary details have been removed and some parts of section 3 have been moved to the appendix and also summarized.

*- p 11, Filtering: A lot of observations are removed due to filtering. This is an important comment. The added value of geostationary satellites lies in their temporal resolution. If you only retain 1 to 4 observations per day, is there still a large added value, is this representative of the future real situation? Please add some discussion*
The filtering (or thinning) procedure that is applied is a spatial filtering. We have found that such thinning procedure did change the assimilated fields slightly only but did save some computing time. A presentation on temporal and spatial representativeness is developed at the beginning of Section 5 and a discussion on its implication for future use of FCI data has been added in Section 6.

*- Location of observations, the observations seem to be concentrated over central Europe, how representative are the results at the AQeR for other regions? You can see that the plots for the entire domain provide different conclusions than the plots for the AQeR stations.*
The results on maps show improvement of scores at most places of the domain. This behaviour is thus consistent with the scores at AQeR stations.

*- Validity of CR-NR: The statistical metrics have been compared to metrics from literature. Two different papers have been used, but if I am correct both evaluating the ensemble of models which is always better than the individual models. Have you also compared to individual model results. The CR-NR seem to be smaller than the NR − real observations difference.*
By introducing differences between the key factors that affect PM forecasts (meteorological drivers, emissions, SOA) and by examining their impact on scores, we aim at evaluating and guaranting sufficient differences between NR and CR.
Besides, since the work of Marécal et al (2015), most of models (and particularly the ones of MACC and CAMS) have improved for PM forecasts and their scores are now in the range of values of the difference between the CR and NR of our study. So it is acceptable that the CR-NR differences are representative of real errors, particularly for CR3 and CR4.

*- Only one observation per hour is used because of the system. As is suggested in the conclusion I would make a super observation. It is mentioned that avoiding overoptimistic results is one of the reasons for this choice but I do not agree as this will probably be done once the observations become available in real life, so I do not see why this would lead to overoptimistic results*
We agree. The comment on the over-optimistic results has been removed (page 12, line 25).

*- Spatially 3 out of 4 observations are removed trough filtering for convergence of cost function, while already a lot of observations are removed due to clouds etc. Is a larger impact of the observations foreseen without this spatial filtering*

It has been verified on a short period that thinning does not change significantly the fields. A comment on this has been added (page 12, lines 30-31). The horizontal correlation length of the background error covariance matrix is set to 0.4° in order to propagate the increments horizontally.

*- p1., line 17 Abstract, change 4-months to 4-month*
Done (page 1, line 17)

*- p2. Line 6. which lead → which leads*
Done (page 2, line 9)

*- p2, line 13, Only the WHO limit values are mentioned, but it would also be good to include the official EU limit values.*

Done (page 2, lines 18-19)

*- p2, line 20, there are more appropriate references for the CAMS services, please add the website and the paper from Marécal which is use further along in the paper.*
Done (page 2, line 25)

*- p2 line 30, MODIS is now also available in a 1x1 km product (MAIAC)*
Done (page 3, line 6)

*- p4, line 10-11 this sentence is unclear to me, do you mean by combining AOD and error characteristics?*
Done (page 4, lines 22-23)

*- p4, line 12, CR, which should represent……(something like the current situation, the situation without use of the observations)*
done (page 4, line 25)

*- p4, line 25-26. Also when you are using two different models, you should evaluate this.*
Done (page 5, line 5)

*- p4, line 27, as MOCAGE is used for both….*
Done (page 5, line 7)

*- p7, line 20, FGE is also used*
Done

*- Figure 2, I find it very hard to see the NR background in central Europe with all the overlying circles and the small plots. It is mentioned that the variability and maxima are well represented, but I cannot evaluate this when I do not see the background.*
The figures have been zoomed in for better clarity.

*- p8, lines 20-24. The underestimation is indeed common, I do think there are many more possible reasons for this, such as underestimation of emissions in cold winter periods, and perhaps the modelling of stable winter conditions with shallow surface layers.*
A sentence has been added (page 8, lines 13-15).

*- P8, line 16: maxima, I would change this word, as I relate maxima to the absolute maximum values, while I think you mean the location of the mamixum values.*
Done (page 8, line 2)

*- p12, line 5 slowest → slower*
Done (page 10, line 24)

*- p13, line 2, especially the CR4: but the bias for CR4 is quite small….*
It is rather CR3, which is more different to the NR than CR4. The text has been changed accordingly (page 11, line 25).

*- p13, line 24, here the purpose of the paper is mentioned, but this should be stated more clearly in the introduction, especially the focus on the continental scale*
done (page 3, line 29; page 4, lines 13-14)

*- p14, line 25-28 Figure 14 versus figure 12, I found it hard to see the improvement, while a large*

*improvement is mentioned, maybe it would be helpful to direct the reader to some specific areas where it is visible. Tables 9 and 10 are clear but only cover central Europe.*
Done (page 13, lines 19-21)

*- p15, lines 5-10 please add here the discussion of AOD-PM relation as suggested in the general Comments*
Done (page 13, lines 30-31), and also in Section 6.

*- p15, line 29-20, what is meant with high spatial and temporal episode?*
Correction done (page 14, line 23)

*- p16, summary, please also add the case where it does not work (simulation 4 at the surface, averaged over whole domain).*

Done (page 17, line 15)

We thanks again the reviewers for their insightful and constructive comments, which have lead to improvements to the manuscript. Besides, we have completed some additional changes regarding format, particularly in the references.

[revised manuscript text omitted]

**Déplacé (insertion) [4]**

**Déplacé (insertion) [3]**

**Déplacé (insertion) [1]**

**Déplacé vers le haut [1]:** Si [... [23]]

**Déplacé (insertion) [2]**

**Déplacé vers le haut [2]:** nde [... [28]]

**Déplacé vers le haut [3]:** $\theta_s$ [... [30]]

**Déplacé vers le haut [4]:** $\rho_g$ [... [32]]

**Déplacé vers le haut [5]:** For [... [33]]

**Déplacé (insertion) [6]**

[revised manuscript text omitted]

Table 9: Bias, RMSE, FGE, Factor of 2, Pearson correlation and Spearman correlation of the AR3LEO simulation taking as reference the NR simulations for hourly PM10 and $PM_{2.5}$ concentrations from January to April 2014. The comparison is made at the same station location as for AQeR stations.

| Page 6 : [1] Supprimé | Descheemaecker | 02/01/2019 11:52:00 |
|---|---|---|

n

| Page 6 : [2] Mis en forme | Descheemaecker | 02/01/2019 11:59:00 |
|---|---|---|

Anglais (États Unis)

| Page 6 : [2] Mis en forme | Descheemaecker | 02/01/2019 11:59:00 |
|---|---|---|

Anglais (États Unis)

| Page 6 : [2] Mis en forme | Descheemaecker | 02/01/2019 11:59:00 |
|---|---|---|

Anglais (États Unis)

| Page 6 : [2] Mis en forme | Descheemaecker | 02/01/2019 11:59:00 |
|---|---|---|

Anglais (États Unis)

| Page 6 : [2] Mis en forme | Descheemaecker | 02/01/2019 11:59:00 |
|---|---|---|

Anglais (États Unis)

| Page 6 : [2] Mis en forme | Descheemaecker | 02/01/2019 11:59:00 |
|---|---|---|

Anglais (États Unis)

| Page 6 : [2] Mis en forme | Descheemaecker | 02/01/2019 11:59:00 |
|---|---|---|

Anglais (États Unis)

| Page 6 : [2] Mis en forme | Descheemaecker | 02/01/2019 11:59:00 |
|---|---|---|

Anglais (États Unis)

| Page 6 : [2] Mis en forme | Descheemaecker | 02/01/2019 11:59:00 |
|---|---|---|

Anglais (États Unis)

| Page 6 : [2] Mis en forme | Descheemaecker | 02/01/2019 11:59:00 |
|---|---|---|

Anglais (États Unis)

| Page 6 : [2] Mis en forme | Descheemaecker | 02/01/2019 11:59:00 |
|---|---|---|

Anglais (États Unis)

| Page 6 : [2] Mis en forme | Descheemaecker | 02/01/2019 11:59:00 |
|---|---|---|

Anglais (États Unis)

| Page 6 : [2] Mis en forme | Descheemaecker | 02/01/2019 11:59:00 |
|---|---|---|

Anglais (États Unis)

| Page 6 : [2] Mis en forme | Descheemaecker | 02/01/2019 11:59:00 |
|---|---|---|

Anglais (États Unis)

**Page 6 : [2] Mis en forme**        **Descheemaecker**        **02/01/2019 11:59:00**
Anglais (États Unis)

**Page 6 : [2] Mis en forme**        **Descheemaecker**        **02/01/2019 11:59:00**
Anglais (États Unis)

**Page 6 : [2] Mis en forme**        **Descheemaecker**        **02/01/2019 11:59:00**
Anglais (États Unis)

**Page 6 : [2] Mis en forme**        **Descheemaecker**        **02/01/2019 11:59:00**
Anglais (États Unis)

**Page 6 : [2] Mis en forme**        **Descheemaecker**        **02/01/2019 11:59:00**
Anglais (États Unis)

**Page 6 : [2] Mis en forme**        **Descheemaecker**        **02/01/2019 11:59:00**
Anglais (États Unis)

**Page 6 : [2] Mis en forme**        **Descheemaecker**        **02/01/2019 11:59:00**
Anglais (États Unis)

**Page 6 : [2] Mis en forme**        **Descheemaecker**        **02/01/2019 11:59:00**
Anglais (États Unis)

**Page 6 : [2] Mis en forme**        **Descheemaecker**        **02/01/2019 11:59:00**
Anglais (États Unis)

**Page 6 : [2] Mis en forme**        **Descheemaecker**        **02/01/2019 11:59:00**
Anglais (États Unis)

**Page 6 : [2] Mis en forme**        **Descheemaecker**        **02/01/2019 11:59:00**
Anglais (États Unis)

**Page 6 : [2] Mis en forme**        **Descheemaecker**        **02/01/2019 11:59:00**
Anglais (États Unis)

**Page 6 : [2] Mis en forme**        **Descheemaecker**        **02/01/2019 11:59:00**
Anglais (États Unis)

**Page 6 : [2] Mis en forme**        **Descheemaecker**        **02/01/2019 11:59:00**
Anglais (États Unis)

**Page 6 : [2] Mis en forme**        **Descheemaecker**        **02/01/2019 11:59:00**
Anglais (États Unis)

**Page 6 : [2] Mis en forme**        **Descheemaecker**        **02/01/2019 11:59:00**
Anglais (États Unis)

**Page 6 : [2] Mis en forme**        **Descheemaecker**        **02/01/2019 11:59:00**
Anglais (États Unis)

| Page 6 : [2] Mis en forme | Descheemaecker | 02/01/2019 11:59:00 |

Anglais (États Unis)

| Page 6 : [2] Mis en forme | Descheemaecker | 02/01/2019 11:59:00 |

Anglais (États Unis)

| Page 6 : [2] Mis en forme | Descheemaecker | 02/01/2019 11:59:00 |

Anglais (États Unis)

| Page 6 : [2] Mis en forme | Descheemaecker | 02/01/2019 11:59:00 |

Anglais (États Unis)

| Page 6 : [3] Supprimé | Descheemaecker | 10/12/2018 08:40:00 |

is computation is based on

| Page 6 : [3] Supprimé | Descheemaecker | 10/12/2018 08:40:00 |

is computation is based on

| Page 6 : [3] Supprimé | Descheemaecker | 10/12/2018 08:40:00 |

is computation is based on

| Page 6 : [3] Supprimé | Descheemaecker | 10/12/2018 08:40:00 |

is computation is based on

| Page 6 : [3] Supprimé | Descheemaecker | 10/12/2018 08:40:00 |

is computation is based on

| Page 7 : [4] Supprimé | Descheemaecker | 02/01/2019 10:12:00 |

; Diehl et al., 2012

| Page 7 : [4] Supprimé | Descheemaecker | 02/01/2019 10:12:00 |

; Diehl et al., 2012

| Page 7 : [5] Supprimé | Descheemaecker | 03/12/2018 09:44:00 |

several

| Page 7 : [5] Supprimé | Descheemaecker | 03/12/2018 09:44:00 |

several

| Page 7 : [5] Supprimé | Descheemaecker | 03/12/2018 09:44:00 |

several

| Page 7 : [5] Supprimé | Descheemaecker | 03/12/2018 09:44:00 |

several

| Page 7 : [5] Supprimé | Descheemaecker | 03/12/2018 09:44:00 |

several

| Page 7 : [6] Mis en forme | Descheemaecker | 02/01/2019 11:59:00 |

Anglais (États Unis)

| Page 7 : [6] Mis en forme | Descheemaecker | 02/01/2019 11:59:00 |

Anglais (États Unis)

| Page 7 : [6] Mis en forme | Descheemaecker | 02/01/2019 11:59:00 |
|---|---|---|

Anglais (États Unis)

| Page 7 : [6] Mis en forme | Descheemaecker | 02/01/2019 11:59:00 |
|---|---|---|

Anglais (États Unis)

| Page 7 : [6] Mis en forme | Descheemaecker | 02/01/2019 11:59:00 |
|---|---|---|

Anglais (États Unis)

| Page 7 : [6] Mis en forme | Descheemaecker | 02/01/2019 11:59:00 |
|---|---|---|

Anglais (États Unis)

| Page 7 : [6] Mis en forme | Descheemaecker | 02/01/2019 11:59:00 |
|---|---|---|

Anglais (États Unis)

| Page 7 : [6] Mis en forme | Descheemaecker | 02/01/2019 11:59:00 |
|---|---|---|

Anglais (États Unis)

| Page 7 : [6] Mis en forme | Descheemaecker | 02/01/2019 11:59:00 |
|---|---|---|

Anglais (États Unis)

| Page 7 : [6] Mis en forme | Descheemaecker | 02/01/2019 11:59:00 |
|---|---|---|

Anglais (États Unis)

| Page 7 : [6] Mis en forme | Descheemaecker | 02/01/2019 11:59:00 |
|---|---|---|

Anglais (États Unis)

| Page 7 : [6] Mis en forme | Descheemaecker | 02/01/2019 11:59:00 |
|---|---|---|

Anglais (États Unis)

| Page 7 : [6] Mis en forme | Descheemaecker | 02/01/2019 11:59:00 |
|---|---|---|

Anglais (États Unis)

| Page 7 : [6] Mis en forme | Descheemaecker | 02/01/2019 11:59:00 |
|---|---|---|

Anglais (États Unis)

| Page 7 : [6] Mis en forme | Descheemaecker | 02/01/2019 11:59:00 |
|---|---|---|

Anglais (États Unis)

| Page 7 : [6] Mis en forme | Descheemaecker | 02/01/2019 11:59:00 |
|---|---|---|

Anglais (États Unis)

| Page 7 : [6] Mis en forme | Descheemaecker | 02/01/2019 11:59:00 |
|---|---|---|

Anglais (États Unis)

| Page 7 : [6] Mis en forme | Descheemaecker | 02/01/2019 11:59:00 |
|---|---|---|

Anglais (États Unis)

**Page 7 : [6] Mis en forme**        **Descheemaecker**        **02/01/2019 11:59:00**

Anglais (États Unis)

**Page 7 : [6] Mis en forme**        **Descheemaecker**        **02/01/2019 11:59:00**

Anglais (États Unis)

**Page 7 : [6] Mis en forme**        **Descheemaecker**        **02/01/2019 11:59:00**

Anglais (États Unis)

**Page 7 : [7] Mis en forme**        **Descheemaecker**        **02/01/2019 11:59:00**

Anglais (États Unis)

**Page 7 : [7] Mis en forme**        **Descheemaecker**        **02/01/2019 11:59:00**

Anglais (États Unis)

**Page 7 : [7] Mis en forme**        **Descheemaecker**        **02/01/2019 11:59:00**

Anglais (États Unis)

**Page 7 : [7] Mis en forme**        **Descheemaecker**        **02/01/2019 11:59:00**

Anglais (États Unis)

**Page 7 : [7] Mis en forme**        **Descheemaecker**        **02/01/2019 11:59:00**

Anglais (États Unis)

**Page 7 : [7] Mis en forme**        **Descheemaecker**        **02/01/2019 11:59:00**

Anglais (États Unis)

**Page 7 : [7] Mis en forme**        **Descheemaecker**        **02/01/2019 11:59:00**

Anglais (États Unis)

**Page 7 : [7] Mis en forme**        **Descheemaecker**        **02/01/2019 11:59:00**

Anglais (États Unis)

**Page 7 : [7] Mis en forme**        **Descheemaecker**        **02/01/2019 11:59:00**

Anglais (États Unis)

**Page 7 : [7] Mis en forme**        **Descheemaecker**        **02/01/2019 11:59:00**

Anglais (États Unis)

**Page 7 : [7] Mis en forme**        **Descheemaecker**        **02/01/2019 11:59:00**

Anglais (États Unis)

**Page 7 : [7] Mis en forme**        **Descheemaecker**        **02/01/2019 11:59:00**

Anglais (États Unis)

**Page 7 : [7] Mis en forme**        **Descheemaecker**        **02/01/2019 11:59:00**

Anglais (États Unis)

**Page 7 : [7] Mis en forme**        **Descheemaecker**        **02/01/2019 11:59:00**

Anglais (États Unis)

**Page 7 : [7] Mis en forme**        **Descheemaecker**        **02/01/2019 11:59:00**

Anglais (États Unis)

| Page 7 : [8] Supprimé | Descheemaecker | 03/12/2018 09:44:00 |
|---|---|---|

R

| Page 7 : [8] Supprimé | Descheemaecker | 03/12/2018 09:44:00 |
|---|---|---|

R

| Page 7 : [8] Supprimé | Descheemaecker | 03/12/2018 09:44:00 |
|---|---|---|

R

| Page 7 : [8] Supprimé | Descheemaecker | 03/12/2018 09:44:00 |
|---|---|---|

R

| Page 7 : [8] Supprimé | Descheemaecker | 03/12/2018 09:44:00 |
|---|---|---|

R

| Page 7 : [9] Mis en forme | Descheemaecker | 02/01/2019 11:59:00 |
|---|---|---|

Anglais (États Unis)

| Page 7 : [9] Mis en forme | Descheemaecker | 02/01/2019 11:59:00 |
|---|---|---|

Anglais (États Unis)

| Page 7 : [9] Mis en forme | Descheemaecker | 02/01/2019 11:59:00 |
|---|---|---|

Anglais (États Unis)

| Page 7 : [9] Mis en forme | Descheemaecker | 02/01/2019 11:59:00 |
|---|---|---|

Anglais (États Unis)

| Page 7 : [9] Mis en forme | Descheemaecker | 02/01/2019 11:59:00 |
|---|---|---|

Anglais (États Unis)

| Page 7 : [9] Mis en forme | Descheemaecker | 02/01/2019 11:59:00 |
|---|---|---|

Anglais (États Unis)

| Page 7 : [9] Mis en forme | Descheemaecker | 02/01/2019 11:59:00 |
|---|---|---|

Anglais (États Unis)

| Page 7 : [9] Mis en forme | Descheemaecker | 02/01/2019 11:59:00 |
|---|---|---|

Anglais (États Unis)

| Page 7 : [9] Mis en forme | Descheemaecker | 02/01/2019 11:59:00 |
|---|---|---|

Anglais (États Unis)

| Page 7 : [9] Mis en forme | Descheemaecker | 02/01/2019 11:59:00 |
|---|---|---|

Anglais (États Unis)

| Page 7 : [10] Supprimé | Descheemaecker | 03/12/2018 09:44:00 |
|---|---|---|

the F

| Page 7 : [10] Supprimé | Descheemaecker | 03/12/2018 09:44:00 |
|---|---|---|

the F

| Page 7 : [10] Supprimé | Descheemaecker | 03/12/2018 09:44:00 |
|---|---|---|

the F

| Page 7 : [10] Supprimé | Descheemaecker | 03/12/2018 09:44:00 |
|---|---|---|

the F

| Page 7 : [11] Mis en forme | Descheemaecker | 02/01/2019 11:59:00 |
|---|---|---|

Anglais (États Unis)

| Page 7 : [11] Mis en forme | Descheemaecker | 02/01/2019 11:59:00 |
|---|---|---|

Anglais (États Unis)

| Page 7 : [11] Mis en forme | Descheemaecker | 02/01/2019 11:59:00 |
|---|---|---|

Anglais (États Unis)

| Page 7 : [11] Mis en forme | Descheemaecker | 02/01/2019 11:59:00 |
|---|---|---|

Anglais (États Unis)

| Page 7 : [11] Mis en forme | Descheemaecker | 02/01/2019 11:59:00 |
|---|---|---|

Anglais (États Unis)

| Page 7 : [11] Mis en forme | Descheemaecker | 02/01/2019 11:59:00 |
|---|---|---|

Anglais (États Unis)

| Page 7 : [11] Mis en forme | Descheemaecker | 02/01/2019 11:59:00 |
|---|---|---|

Anglais (États Unis)

| Page 7 : [11] Mis en forme | Descheemaecker | 02/01/2019 11:59:00 |
|---|---|---|

Anglais (États Unis)

| Page 7 : [11] Mis en forme | Descheemaecker | 02/01/2019 11:59:00 |
|---|---|---|

Anglais (États Unis)

| Page 7 : [11] Mis en forme | Descheemaecker | 02/01/2019 11:59:00 |
|---|---|---|

Anglais (États Unis)

| Page 7 : [11] Mis en forme | Descheemaecker | 02/01/2019 11:59:00 |
|---|---|---|

Anglais (États Unis)

| Page 7 : [11] Mis en forme | Descheemaecker | 02/01/2019 11:59:00 |
|---|---|---|

Anglais (États Unis)

| Page 7 : [11] Mis en forme | Descheemaecker | 02/01/2019 11:59:00 |
|---|---|---|

Anglais (États Unis)

| Page 7 : [12] Supprimé | Descheemaecker | 03/12/2018 09:45:00 |
|---|---|---|

The

| Page 7 : [13] Mis en forme | Descheemaecker | 02/01/2019 11:59:00 |
|---|---|---|

Anglais (États Unis)

| Page 7 : [13] Mis en forme | Descheemaecker | 02/01/2019 11:59:00 |
|---|---|---|

Anglais (États Unis)

| Page 7 : [13] Mis en forme | Descheemaecker | 02/01/2019 11:59:00 |
|---|---|---|

Anglais (États Unis)

| Page 7 : [13] Mis en forme | Descheemaecker | 02/01/2019 11:59:00 |
|---|---|---|

Anglais (États Unis)

| Page 7 : [13] Mis en forme | Descheemaecker | 02/01/2019 11:59:00 |
|---|---|---|

Anglais (États Unis)

| Page 7 : [13] Mis en forme | Descheemaecker | 02/01/2019 11:59:00 |
|---|---|---|

Anglais (États Unis)

| Page 7 : [13] Mis en forme | Descheemaecker | 02/01/2019 11:59:00 |
|---|---|---|

Anglais (États Unis)

| Page 7 : [13] Mis en forme | Descheemaecker | 02/01/2019 11:59:00 |
|---|---|---|

Anglais (États Unis)

| Page 7 : [13] Mis en forme | Descheemaecker | 02/01/2019 11:59:00 |
|---|---|---|

Anglais (États Unis)

| Page 7 : [13] Mis en forme | Descheemaecker | 02/01/2019 11:59:00 |
|---|---|---|

Anglais (États Unis)

| Page 7 : [13] Mis en forme | Descheemaecker | 02/01/2019 11:59:00 |
|---|---|---|

Anglais (États Unis)

| Page 7 : [13] Mis en forme | Descheemaecker | 02/01/2019 11:59:00 |
|---|---|---|

Anglais (États Unis)

| Page 7 : [13] Mis en forme | Descheemaecker | 02/01/2019 11:59:00 |
|---|---|---|

Anglais (États Unis)

| Page 7 : [13] Mis en forme | Descheemaecker | 02/01/2019 11:59:00 |
|---|---|---|

Anglais (États Unis)

| Page 7 : [13] Mis en forme | Descheemaecker | 02/01/2019 11:59:00 |
|---|---|---|

Anglais (États Unis)

| Page 7 : [13] Mis en forme | Descheemaecker | 02/01/2019 11:59:00 |
|---|---|---|

Anglais (États Unis)

| Page 7 : [13] Mis en forme | Descheemaecker | 02/01/2019 11:59:00 |
|---|---|---|

Anglais (États Unis)

**Page 7 : [13] Mis en forme** | **Descheemaecker** | **02/01/2019 11:59:00**

Anglais (États Unis)

**Page 7 : [13] Mis en forme** | **Descheemaecker** | **02/01/2019 11:59:00**

Anglais (États Unis)

**Page 7 : [13] Mis en forme** | **Descheemaecker** | **02/01/2019 11:59:00**

Anglais (États Unis)

**Page 7 : [13] Mis en forme** | **Descheemaecker** | **02/01/2019 11:59:00**

Anglais (États Unis)

**Page 7 : [13] Mis en forme** | **Descheemaecker** | **02/01/2019 11:59:00**

Anglais (États Unis)

**Page 7 : [13] Mis en forme** | **Descheemaecker** | **02/01/2019 11:59:00**

Anglais (États Unis)

**Page 7 : [13] Mis en forme** | **Descheemaecker** | **02/01/2019 11:59:00**

Anglais (États Unis)

**Page 7 : [13] Mis en forme** | **Descheemaecker** | **02/01/2019 11:59:00**

Anglais (États Unis)

**Page 7 : [13] Mis en forme** | **Descheemaecker** | **02/01/2019 11:59:00**

Anglais (États Unis)

**Page 7 : [13] Mis en forme** | **Descheemaecker** | **02/01/2019 11:59:00**

Anglais (États Unis)

**Page 7 : [13] Mis en forme** | **Descheemaecker** | **02/01/2019 11:59:00**

Anglais (États Unis)

**Page 7 : [13] Mis en forme** | **Descheemaecker** | **02/01/2019 11:59:00**

Anglais (États Unis)

**Page 7 : [13] Mis en forme** | **Descheemaecker** | **02/01/2019 11:59:00**

Anglais (États Unis)

**Page 7 : [13] Mis en forme** | **Descheemaecker** | **02/01/2019 11:59:00**

Anglais (États Unis)

**Page 7 : [13] Mis en forme** | **Descheemaecker** | **02/01/2019 11:59:00**

Anglais (États Unis)

**Page 7 : [13] Mis en forme** | **Descheemaecker** | **02/01/2019 11:59:00**

Anglais (États Unis)

| Page 7 : [13] Mis en forme | Descheemaecker | 02/01/2019 11:59:00 |

Anglais (États Unis)

| Page 7 : [13] Mis en forme | Descheemaecker | 02/01/2019 11:59:00 |

Anglais (États Unis)

| Page 7 : [13] Mis en forme | Descheemaecker | 02/01/2019 11:59:00 |

Anglais (États Unis)

| Page 7 : [13] Mis en forme | Descheemaecker | 02/01/2019 11:59:00 |

Anglais (États Unis)

| Page 7 : [13] Mis en forme | Descheemaecker | 02/01/2019 11:59:00 |

Anglais (États Unis)

| Page 7 : [13] Mis en forme | Descheemaecker | 02/01/2019 11:59:00 |

Anglais (États Unis)

| Page 7 : [13] Mis en forme | Descheemaecker | 02/01/2019 11:59:00 |

Anglais (États Unis)

| Page 7 : [13] Mis en forme | Descheemaecker | 02/01/2019 11:59:00 |

Anglais (États Unis)

| Page 7 : [13] Mis en forme | Descheemaecker | 02/01/2019 11:59:00 |

Anglais (États Unis)

| Page 7 : [14] Supprimé | Descheemaecker | 03/12/2018 09:48:00 |

a

| Page 7 : [14] Supprimé | Descheemaecker | 03/12/2018 09:48:00 |

a

| Page 7 : [14] Supprimé | Descheemaecker | 03/12/2018 09:48:00 |

a

| Page 7 : [15] Mis en forme | Descheemaecker | 02/01/2019 11:59:00 |

Anglais (États Unis)

| Page 7 : [15] Mis en forme | Descheemaecker | 02/01/2019 11:59:00 |

Anglais (États Unis)

| Page 7 : [15] Mis en forme | Descheemaecker | 02/01/2019 11:59:00 |

Anglais (États Unis)

| Page 7 : [15] Mis en forme | Descheemaecker | 02/01/2019 11:59:00 |

Anglais (États Unis)

| Page 7 : [15] Mis en forme | Descheemaecker | 02/01/2019 11:59:00 |

Anglais (États Unis)

| Page 7 : [15] Mis en forme | Descheemaecker | 02/01/2019 11:59:00 |

Anglais (États Unis)

**Page 7 : [15] Mis en forme**      **Descheemaecker**      **02/01/2019 11:59:00**
Anglais (États Unis)

**Page 7 : [15] Mis en forme**      **Descheemaecker**      **02/01/2019 11:59:00**
Anglais (États Unis)

**Page 7 : [15] Mis en forme**      **Descheemaecker**      **02/01/2019 11:59:00**
Anglais (États Unis)

**Page 7 : [15] Mis en forme**      **Descheemaecker**      **02/01/2019 11:59:00**
Anglais (États Unis)

**Page 7 : [15] Mis en forme**      **Descheemaecker**      **02/01/2019 11:59:00**
Anglais (États Unis)

**Page 7 : [15] Mis en forme**      **Descheemaecker**      **02/01/2019 11:59:00**
Anglais (États Unis)

**Page 7 : [15] Mis en forme**      **Descheemaecker**      **02/01/2019 11:59:00**
Anglais (États Unis)

**Page 7 : [15] Mis en forme**      **Descheemaecker**      **02/01/2019 11:59:00**
Anglais (États Unis)

**Page 7 : [15] Mis en forme**      **Descheemaecker**      **02/01/2019 11:59:00**
Anglais (États Unis)

**Page 7 : [15] Mis en forme**      **Descheemaecker**      **02/01/2019 11:59:00**
Anglais (États Unis)

**Page 7 : [15] Mis en forme**      **Descheemaecker**      **02/01/2019 11:59:00**
Anglais (États Unis)

**Page 7 : [15] Mis en forme**      **Descheemaecker**      **02/01/2019 11:59:00**
Anglais (États Unis)

**Page 7 : [15] Mis en forme**      **Descheemaecker**      **02/01/2019 11:59:00**
Anglais (États Unis)

**Page 7 : [15] Mis en forme**      **Descheemaecker**      **02/01/2019 11:59:00**
Anglais (États Unis)

**Page 7 : [15] Mis en forme**      **Descheemaecker**      **02/01/2019 11:59:00**
Anglais (États Unis)

**Page 7 : [15] Mis en forme**      **Descheemaecker**      **02/01/2019 11:59:00**
Anglais (États Unis)

**Page 7 : [15] Mis en forme**      **Descheemaecker**      **02/01/2019 11:59:00**
Anglais (États Unis)

| Page 7 : [15] Mis en forme | Descheemaecker | 02/01/2019 11:59:00 |

Anglais (États Unis)

| Page 7 : [16] Supprimé | Descheemaecker | 05/12/2018 09:52:00 |

 Stations, classified as 1 to 5, have been selected following Joly and Peuch (2012). This selection keeps stations that are representative of background air pollution, which is the range of scale that the model and the satellite may represent.

| Page 9 : [17] Supprimé | Descheemaecker | 03/12/2018 09:37:00 |

S

| Page 9 : [18] Supprimé | Descheemaecker | 31/10/2018 09:06:00 |

from the 3D fields of the NR variables: primary organic and black carbon concentrations, desert dust concentrations, sea salt concentrations, secondary inorganic aerosol concentrations (ammonium, nitrate and sulfate), relative humidity, temperature and pressure. From these fields, AODs are calculated for the FCI VIS04 channel with the same computation module for aerosol optical properties as the observation operator. An AOD error is introduced using characterize errors of the FCI. To characterize the error in the channel VIS04, the simulator developed by Aoun (2016, Aoun et al., 2015), based on the Radiative Transfer Model (RTM) libRadtran (Mayer and Killings, 2005), has been used.

simulator developed by Aoun (2016, Aoun et al., 2015), based on the Radiative Transfer Model (RTM) libRadtran (Mayer and Killings, 2005), has been used.
* * *
**Page 9 : [18] Supprimé**  **Descheemaecker**  **31/10/2018 09:06:00**

from the 3D fields of the NR variables: primary organic and black carbon concentrations, desert dust concentrations, sea salt concentrations, secondary inorganic aerosol concentrations (ammonium, nitrate and sulfate), relative humidity, temperature and pressure. From these fields, AODs are calculated for the FCI VIS04 channel with the same computation module for aerosol optical properties as the observation operator. An AOD error is introduced using characterize errors of the FCI. To characterize the error in the channel VIS04, the simulator developed by Aoun (2016, Aoun et al., 2015), based on the Radiative Transfer Model (RTM) libRadtran (Mayer and Killings, 2005), has been used.
* * ** * *
**Page 9 : [19] Supprimé**  **Descheemaecker**  **02/01/2019 14:01:00**

u
* * ** * *
**Page 9 : [20] Mis en forme**  **Descheemaecker**  **02/01/2019 11:59:00**

Anglais (États Unis)
* * ** * *
**Page 9 : [21] Mis en forme**  **Descheemaecker**  **02/01/2019 11:59:00**

Anglais (États Unis)

| Page 9 : [21] Mis en forme | Descheemaecker | 02/01/2019 11:59:00 |
|---|---|---|

Anglais (États Unis)

| Page 9 : [22] Supprimé | Descheemaecker | 31/10/2018 09:52:00 |
|---|---|---|

It

| Page 9 : [22] Supprimé | Descheemaecker | 31/10/2018 09:52:00 |
|---|---|---|

It

| Page 9 : [22] Supprimé | Descheemaecker | 31/10/2018 09:52:00 |
|---|---|---|

It

[revised manuscript text omitted]

Anglais (États Unis)

Anglais (États Unis)

| | | |
|---|---|---|
| **Page 9 : [30] Mis en forme** | **Descheemaecker** | **02/01/2019 11:59:00** |

Anglais (États Unis)

| | | |
|---|---|---|
| **Page 9 : [30] Mis en forme** | **Descheemaecker** | **02/01/2019 11:59:00** |

Anglais (États Unis)

| | | |
|---|---|---|
| **Page 9 : [30] Mis en forme** | **Descheemaecker** | **02/01/2019 11:59:00** |

Anglais (États Unis)

| | | |
|---|---|---|
| **Page 9 : [30] Mis en forme** | **Descheemaecker** | **02/01/2019 11:59:00** |

Anglais (États Unis)

| | | |
|---|---|---|
| **Page 9 : [30] Mis en forme** | **Descheemaecker** | **02/01/2019 11:59:00** |

Anglais (États Unis)

| | | |
|---|---|---|
| **Page 9 : [30] Mis en forme** | **Descheemaecker** | **02/01/2019 11:59:00** |

Anglais (États Unis)

| | | |
|---|---|---|
| **Page 9 : [30] Mis en forme** | **Descheemaecker** | **02/01/2019 11:59:00** |

Anglais (États Unis)

| | | |
|---|---|---|
| **Page 9 : [30] Mis en forme** | **Descheemaecker** | **02/01/2019 11:59:00** |

Anglais (États Unis)

| | | |
|---|---|---|
| **Page 9 : [30] Mis en forme** | **Descheemaecker** | **02/01/2019 11:59:00** |

Anglais (États Unis)

| | | |
|---|---|---|
| **Page 9 : [30] Mis en forme** | **Descheemaecker** | **02/01/2019 11:59:00** |

Anglais (États Unis)

| | | |
|---|---|---|
| **Page 9 : [30] Mis en forme** | **Descheemaecker** | **02/01/2019 11:59:00** |

Anglais (États Unis)

| | | |
|---|---|---|
| **Page 9 : [30] Mis en forme** | **Descheemaecker** | **02/01/2019 11:59:00** |

Anglais (États Unis)

| | | |
|---|---|---|
| **Page 9 : [30] Mis en forme** | **Descheemaecker** | **02/01/2019 11:59:00** |

Anglais (États Unis)

| | | |
|---|---|---|
| **Page 9 : [30] Mis en forme** | **Descheemaecker** | **02/01/2019 11:59:00** |

Anglais (États Unis)

| | | |
|---|---|---|
| **Page 9 : [30] Mis en forme** | **Descheemaecker** | **02/01/2019 11:59:00** |

Anglais (États Unis)

| | | |
|---|---|---|
| **Page 9 : [30] Mis en forme** | **Descheemaecker** | **02/01/2019 11:59:00** |

Anglais (États Unis)

| | | |
|---|---|---|
| **Page 9 : [30] Mis en forme** | **Descheemaecker** | **02/01/2019 11:59:00** |

Anglais (États Unis)

| Page 9 : [30] Mis en forme | Descheemaecker | 02/01/2019 11:59:00 |
|---|---|---|

Anglais (États Unis)

| Page 9 : [30] Mis en forme | Descheemaecker | 02/01/2019 11:59:00 |
|---|---|---|

Anglais (États Unis)

| Page 9 : [30] Mis en forme | Descheemaecker | 02/01/2019 11:59:00 |
|---|---|---|

Anglais (États Unis)

| Page 9 : [31] Mis en forme | Descheemaecker | 02/01/2019 11:59:00 |
|---|---|---|

Anglais (États Unis)

| Page 9 : [31] Mis en forme | Descheemaecker | 02/01/2019 11:59:00 |
|---|---|---|

Anglais (États Unis)

| Page 9 : [31] Mis en forme | Descheemaecker | 02/01/2019 11:59:00 |
|---|---|---|

Anglais (États Unis)

| Page 9 : [32] Mis en forme | Descheemaecker | 02/01/2019 11:59:00 |
|---|---|---|

Anglais (États Unis)

| Page 9 : [32] Mis en forme | Descheemaecker | 02/01/2019 11:59:00 |
|---|---|---|

Anglais (États Unis)

| Page 9 : [32] Mis en forme | Descheemaecker | 02/01/2019 11:59:00 |
|---|---|---|

Anglais (États Unis)

| Page 9 : [33] Déplacé vers la page 9 (Déplacement n°5) 05/11/2018 15:18:00 | Descheemaecker |
|---|---|

For all OPAC groups, the dependence of reflectance for VIS04 on the total ozone column and water vapour is negligible and is not taken into account in the reflectance approximation.

| Page 9 : [34] Mis en forme | Descheemaecker | 02/01/2019 11:59:00 |
|---|---|---|

Anglais (États Unis)

| Page 9 : [34] Mis en forme | Descheemaecker | 02/01/2019 11:59:00 |
|---|---|---|

Anglais (États Unis)

| Page 9 : [34] Mis en forme | Descheemaecker | 02/01/2019 11:59:00 |
|---|---|---|

Anglais (États Unis)

| Page 9 : [34] Mis en forme | Descheemaecker | 02/01/2019 11:59:00 |
|---|---|---|

Anglais (États Unis)

| Page 9 : [34] Mis en forme | Descheemaecker | 02/01/2019 11:59:00 |
|---|---|---|

Anglais (États Unis)

| Page 9 : [34] Mis en forme | Descheemaecker | 02/01/2019 11:59:00 |
|---|---|---|

Anglais (États Unis)

| Page 9 : [34] Mis en forme | Descheemaecker | 02/01/2019 11:59:00 |
|---|---|---|

Anglais (États Unis)

| Page 9 : [34] Mis en forme | Descheemaecker | 02/01/2019 11:59:00 |

Anglais (États Unis)

| Page 9 : [35] Supprimé | Descheemaecker | 05/11/2018 15:48:00 |

(Table 5) using

| Page 9 : [36] Supprimé | Descheemaecker | 03/12/2018 14:57:00 |

In order to keep a large number of profiles, the criteria are less restrictive than the ones described in Table 4. By this process

| Page 18 : [37] Mis en forme | Descheemaecker | 02/01/2019 11:59:00 |

Police :Gras, Anglais (États Unis)

| Page 18 : [38] Mis en forme | Descheemaecker | 02/01/2019 11:59:00 |

Anglais (États Unis)

| Page 18 : [39] Mis en forme | Descheemaecker | 02/01/2019 11:59:00 |

Anglais (États Unis)

| Page 18 : [39] Mis en forme | Descheemaecker | 02/01/2019 11:59:00 |
|---|---|---|

Anglais (États Unis)

| Page 18 : [39] Mis en forme | Descheemaecker | 02/01/2019 11:59:00 |
|---|---|---|

Anglais (États Unis)

| Page 18 : [40] Mis en forme | Descheemaecker | 02/01/2019 11:59:00 |
|---|---|---|

Anglais (États Unis)

| Page 18 : [40] Mis en forme | Descheemaecker | 02/01/2019 11:59:00 |
|---|---|---|

Anglais (États Unis)

| Page 18 : [40] Mis en forme | Descheemaecker | 02/01/2019 11:59:00 |
|---|---|---|

Anglais (États Unis)

| Page 18 : [40] Mis en forme | Descheemaecker | 02/01/2019 11:59:00 |
|---|---|---|

Anglais (États Unis)

| Page 18 : [40] Mis en forme | Descheemaecker | 02/01/2019 11:59:00 |
|---|---|---|

Anglais (États Unis)

| Page 18 : [41] Mis en forme | Descheemaecker | 02/01/2019 11:59:00 |
|---|---|---|

Anglais (États Unis)

| Page 18 : [41] Mis en forme | Descheemaecker | 02/01/2019 11:59:00 |
|---|---|---|

Anglais (États Unis)

| Page 18 : [41] Mis en forme | Descheemaecker | 02/01/2019 11:59:00 |
|---|---|---|

Anglais (États Unis)

| Page 18 : [41] Mis en forme | Descheemaecker | 02/01/2019 11:59:00 |
|---|---|---|

Anglais (États Unis)

| Page 18 : [41] Mis en forme | Descheemaecker | 02/01/2019 11:59:00 |
|---|---|---|

Anglais (États Unis)

| Page 18 : [41] Mis en forme | Descheemaecker | 02/01/2019 11:59:00 |
|---|---|---|

Anglais (États Unis)

| Page 18 : [41] Mis en forme | Descheemaecker | 02/01/2019 11:59:00 |
|---|---|---|

Anglais (États Unis)

| Page 18 : [41] Mis en forme | Descheemaecker | 02/01/2019 11:59:00 |
|---|---|---|

Anglais (États Unis)

| Page 18 : [41] Mis en forme | Descheemaecker | 02/01/2019 11:59:00 |
|---|---|---|

Anglais (États Unis)

| Page 18 : [41] Mis en forme | Descheemaecker | 02/01/2019 11:59:00 |
|---|---|---|

Anglais (États Unis)

| Page 18 : [42] Mis en forme | Descheemaecker | 02/01/2019 11:59:00 |
|---|---|---|

Anglais (États Unis)

| Page 18 : [43] Mis en forme | Descheemaecker | 02/01/2019 11:59:00 |
|---|---|---|

Anglais (États Unis)

| Page 18 : [44] Mis en forme | Descheemaecker | 02/01/2019 11:59:00 |
|---|---|---|

Anglais (États Unis)

| Page 18 : [45] Mis en forme | Descheemaecker | 02/01/2019 11:59:00 |
|---|---|---|

Anglais (États Unis)

| Page 18 : [46] Mis en forme | Descheemaecker | 02/01/2019 11:59:00 |
|---|---|---|

Anglais (États Unis)

| Page 18 : [47] Mis en forme | Descheemaecker | 02/01/2019 11:59:00 |
|---|---|---|

Anglais (États Unis)

| Page 18 : [47] Mis en forme | Descheemaecker | 02/01/2019 11:59:00 |
|---|---|---|

Anglais (États Unis)

| Page 18 : [47] Mis en forme | Descheemaecker | 02/01/2019 11:59:00 |
|---|---|---|

Anglais (États Unis)

| Page 18 : [47] Mis en forme | Descheemaecker | 02/01/2019 11:59:00 |
|---|---|---|

Anglais (États Unis)

| Page 18 : [47] Mis en forme | Descheemaecker | 02/01/2019 11:59:00 |
|---|---|---|

Anglais (États Unis)

| Page 18 : [47] Mis en forme | Descheemaecker | 02/01/2019 11:59:00 |
|---|---|---|

Anglais (États Unis)

| Page 18 : [47] Mis en forme | Descheemaecker | 02/01/2019 11:59:00 |
|---|---|---|

Anglais (États Unis)

| Page 18 : [48] Mis en forme | Descheemaecker | 02/01/2019 11:59:00 |
|---|---|---|

Anglais (États Unis)

| Page 18 : [48] Mis en forme | Descheemaecker | 02/01/2019 11:59:00 |
|---|---|---|

Anglais (États Unis)

| Page 18 : [48] Mis en forme | Descheemaecker | 02/01/2019 11:59:00 |
|---|---|---|

Anglais (États Unis)

| Page 18 : [49] Mis en forme | Descheemaecker | 02/01/2019 11:59:00 |
|---|---|---|

Anglais (États Unis)

| Page 18 : [49] Mis en forme | Descheemaecker | 02/01/2019 11:59:00 |
|---|---|---|

Anglais (États Unis)

| Page 18 : [49] Mis en forme | Descheemaecker | 02/01/2019 11:59:00 |
|---|---|---|

Anglais (États Unis)

[revised manuscript text omitted]